# Tracing the source of nitrate in a forested stream showing elevated concentrations during storm events

Weitian Ding[1], Urumu Tsunogai[1], Fumiko Nakagawa[1], Takashi Sambuichi[1], Hiroyuki

Sase[2], Masayuki Morohashi[2], Hiroki Yotsuyanagi[2]

[1] Graduate School of Environmental Studies, Nagoya University, Furo-cho, Chikusa-

ku, Nagoya 464-8601, Japan

[2]Asia Center for Air Pollution Research, 1182 Sowa, Nishi-ku, Niigata-shi, Niigata 950-

2144, Japan

Corresponding author: Weitian Ding, Email: ding.weitian.v2@s.mail.nagoya-u.ac.jp

**Abstract**

2        To clarify the source of nitrate increased during storm events in a temperate forested

stream, we monitored temporal variation in the concentrations and stable isotopic
compositions including $\Delta^{17}O$ of stream nitrate in a forested catchment (KJ catchment,
Japan) during three storm events I, II, and III (summer). The stream showed significant
increase in nitrate concentration, from 24.7 μM to 122.6 μM, from 28.7 μM to 134.1
μM, and from 46.6 μM to 114.5 μM during the storm events I, II, and III, respectively.
On the other hand, the isotopic compositions ($\delta^{15}N$, $\delta^{18}O$, and $\Delta^{17}O$) of stream nitrate
showed a decrease in accordance with the increase in the stream nitrate concentration,
from +2.5 ‰ to −0.1 ‰, from +3.0 ‰ to −0.5 ‰, and from +3.5 ‰ to −0.1 ‰ for $\delta^{15}N$,
from +3.1 ‰ to −3.4 ‰, from +2.9 ‰ to −2.5 ‰, and from +2.1 ‰ to −2.3 ‰ for $\delta^{18}O$,
and from +1.6 ‰ to +0.3 ‰, from +1.4 ‰ to +0.3 ‰, and from +1.2 ‰ to +0.5 ‰ for
$\Delta^{17}O$ during the storm events I, II, and III, respectively. Besides, we found strong linear
relationships between the isotopic compositions of stream nitrate and the reciprocal of
stream nitrate concentrations during each storm event, implying that the temporal
variation in the stream nitrate can be explained by simple mixing between two
distinctive endmembers of nitrate having different isotopic compositions. Furthermore,
we found that both concentrations and the isotopic compositions of soil nitrate obtained
in the riparian zone of the stream were plotted on the nitrate-enriched extension of the
linear relationship. We conclude that the soil nitrate in the riparian zone was primarily
responsible for the increase in stream nitrate during the storm events. In addition, we

found that the concentration of unprocessed atmospheric nitrate in the stream was stable at $1.6 \pm 0.4$ µM, $1.8 \pm 0.4$ µM, and $2.1 \pm 0.4$ µM during the storm events I, II, and III, respectively, irrespective to the significant variations in the total nitrate concentration. We conclude that the storm events have little impacts on the concentration of unprocessed atmospheric nitrate in the stream and thus the annual export flux of unprocessed atmospheric nitrate relative to the annual deposition flux can be a robust index to evaluate nitrogen saturation in forested catchments, irrespective to the variation in the number of storm events and/or the variation in the elapsed time from storm events to sampling.

**1 Introduction**

Nitrate is an important nitrogenous nutrient in biosphere. Traditionally, forested ecosystems have been considered nitrogen limited (Vitousek and Howarth, 1991). Due to the elevated loading of nitrogen through atmospheric deposition, however, some forested ecosystems become nitrogen saturated (Aber et al., 1989), from which elevated levels of nitrate are exported (Mitchell et al., 1997; Peterjohn et al., 1996). In addition, sudden increase in the concentration of nitrate in response to storm events has been reported in forested streams worldwide (Aguilera and Melack, 2018; Creed et al., 1996; Kamisako et al., 2008; McHale et al., 2002), which further enhanced nitrate export from forested ecosystems.

Such excessive leaching of nitrate from forested catchment degrades water quality

and cause eutrophication in downstream areas (Galloway et al., 2003; Paerl and
Huisman, 2009). Thus, tracing the source of nitrate increase during storm events in
forested streams is important for sustainable forest management, especially for the
nitrogen-saturated forested ecosystems.
As for the source of nitrate that was added to stream during storm events, either of
the two possible sources have been assumed in past studies; (1) atmospheric nitrate
($NO_3^-{}_{atm}$) in rainwater originally and being supplied directly to stream water (Inamdar
and Mitchell, 2006), and (2) soil nitrate originally and being supplied to stream water
by the flushing effects on soils (Creed et al., 1996; Ocampo et al., 2006). Nevertheless,
monitoring the variation in nitrate concentration, it is difficult to clarify the primary
source of nitrate that increases during storm events.
The natural stable isotopic composition of nitrate has been widely applied to clarify
the sources of nitrate in natural freshwater systems (Burns and Kendall, 2002; Durka et
al., 1994; Kendall et al., 2007). In particular, triple oxygen isotopic compositions of
nitrate ($\Delta^{17}O$) have been used in recent days as a conservative tracer of $NO_3^-{}_{atm}$
deposited onto a forested catchment (Inoue et al., 2021; Michalski et al., 2004;
Nakagawa et al., 2018; Tsunogai et al., 2014), showing distinctively different $\Delta^{17}O$ from
that of remineralized nitrate ($NO_3^-{}_{re}$), derived from organic nitrogen through general
chemical reactions, including microbial N mineralization and microbial nitrification.
While $NO_3^-{}_{re}$, the oxygen atoms of which are derived from either terrestrial $O_2$ or $H_2O$
through microbial processing (i.e., nitrification), always shows the relation close to the
"mass-dependent" relative relation between $^{17}O/^{16}O$ ratios and $^{18}O/^{16}O$ ratios; $NO_3^-{}_{atm}$
displays an anomalous enrichment in $^{17}O$ reflecting oxygen atom transfers from
atmospheric ozone ($O_3$) during the conversion of $NO_X$ to $NO_3^-{}_{atm}$ (Alexander et al.,
2009; Michalski et al., 2003; Morin et al., 2011; Nelson et al., 2018). As a result, the
$\Delta^{17}O$ signature defined by the following equation (Kaiser et al., 2007) enables us to
distinguish $NO_3^-{}_{atm}$ ($\Delta^{17}O > 0$) from $NO_3^-{}_{re}$ ($\Delta^{17}O = 0$):
$$\Delta^{17}O = \frac{1 + \delta^{17}O}{(1 + \delta^{18}O)^\beta} - 1 \qquad (1)$$
where the constant $\beta$ is 0.5279 (Kaiser et al., 2007), $\delta^{18}O = R_{sample}/R_{standard} - 1$ and $R$ is
the $^{18}O/^{16}O$ ratio (or the $^{17}O/^{16}O$ ratio in the case of $\delta^{17}O$ or the $^{15}N/^{14}N$ ratio in the case
of $\delta^{15}N$) of the sample and each standard reference material. In addition, $\Delta^{17}O$ is almost
stable during "mass-dependent" isotope fractionation processes within terrestrial
ecosystems. Therefore, while the $\delta^{15}N$ or $\delta^{18}O$ signature of $NO_3^-{}_{atm}$ can be overprinted
by the biological processes subsequent to deposition, $\Delta^{17}O$ can be used as a robust tracer
of unprocessed $NO_3^-{}_{atm}$ to reflect its accurate mole fraction within total $NO_3^-$, regardless
of the progress of the partial metabolism (partial removal of nitrate through
denitrification and assimilation) subsequent to deposition (Michalski et al., 2004;
Nakagawa et al., 2013, 2018; Tsunogai et al., 2011, 2014, 2018).

81        While the variation in the $\delta^{18}O$ and/or $\Delta^{17}O$ of nitrate in forested streams during storm

events have been reported in past studies (Sebestyen et al., 2019; Sabo et al., 2016;
Buda and Dewalle. 2009), the temporal resolutions of sampling were less than 10
times/day during storm events and the source of the stream nitrate increased during
storm events has not been clarified yet. In this study, we determined the temporal
variation in the concentrations and the isotopic compositions ($\delta^{15}N$, $\delta^{18}O$, and $\Delta^{17}O$) of
stream nitrate at once every hour during storm events in a forested catchment to clarify
(1) the source of nitrate in a forested stream that was added during storm events, and
(2) the temporal variation in the concentration of $NO_3^-{}_{atm}$ in response to storm events.
In addition, the impacts of storm events on the index of nitrogen saturation lately
proposed by Nakagawa et al. (2018) were discussed.

**2 Methods**
2.1 Study site
As for the studying field to trace the source of stream nitrate during storm events, we
chose Kajikawa forested catchment (KJ catchment) in Japan, in which several past
studies had been done to clarify the temporal variation in the concentration of stream
nitrate and the status of nitrogen saturation (Kamisako et al., 2008; Nakagawa et al.,
2018; Sase et al., 2021). This is a small, forested catchment (3.84 ha) located in the
northern part of Shibata City, Niigata Prefecture, along the coast of Sea of Japan (Fig.
1a). The KJ catchment predominantly slopes towards the west-northwest, with a mean
slope of 36°, and the elevation ranges from 60 to 170 m above sea level (Fig. 1b). The
catchment is fully covered by Japanese cedars (*Cryptomeria japonica* D. Don) that were
approximately 46 years old in 2018 (Sase et al., 2021). The parent material is
granodiorite and brown forest soils (Cambisols) have developed in this area (Kamisako
et al., 2008; Sase et al., 2008). The lowest, highest, and mean monthly temperatures
recorded at the nearest meteorological station (Nakajo station) were 1.0 °C (in February),
27.9 °C (in August), and 14.5 °C, respectively, from 2017/5 to 2020/3. The annual mean
precipitation was around 2500 mm, approximately 17% of which occurred during
spring (from March to May), approximately 20% during summer (from June to August),
approximately 28% during fall (from September to November), and approximately 35%
during winter (from December to February). The catchment usually experiences
snowfall from late December to March.
From 2003 to 2005, Kamisako et al. (2008) determined temporal variation in the
concentration of $Ca^{2+}$, $Mg^{2+}$, $Cl^-$, and $NO_3^-$ eluted from the catchment via a stream at
intervals of 1 to 3 hour for 2 to 3 days on each and found that significant increase in the
stream nitrate concentration during storm events, from less than 30 μM to more than
120 μM. On the basis of the observed nitrate enrichment in the stream water, they
concluded that atmospheric nitrogen inputs exceeded the biological demand at the
catchment and proposed that the KJ catchment was under nitrogen saturation.
Nakagawa et al. (2018) determined temporal variation in the concentrations and stable
isotopic compositions ($\delta^{15}N$, $\delta^{18}O$, and $\Delta^{17}O$) of both stream nitrate and soil nitrate for
two years (from 2012/12 to 2014/12) and concluded that nitrate in the groundwater of
the catchment was the major source of nitrate in the stream water during the base flow
periods. Additionally, Nakagawa et al. (2018), who proposed the export flux of $NO_3^-{}_{atm}$
($M_{atm}$) relative to the deposition flux of $NO_3^-{}_{atm}$ ($D_{atm}$) can be an alternative, more robust
index for nitrogen saturation in temperate forested catchments, clarified that the
$M_{atm}/D_{atm}$ ratio in the KJ catchment was larger (9.4 %) than the other catchments they
studied simultaneously (6.5 % and 2.6 %), which also implied the KJ catchment was
under the nitrogen saturation. Moreover, Sase et al. (2021) reported the
nitrate concentration of the stream has been increasing in recent years, which implies
that nitrogen saturation is still ongoing in the forest.

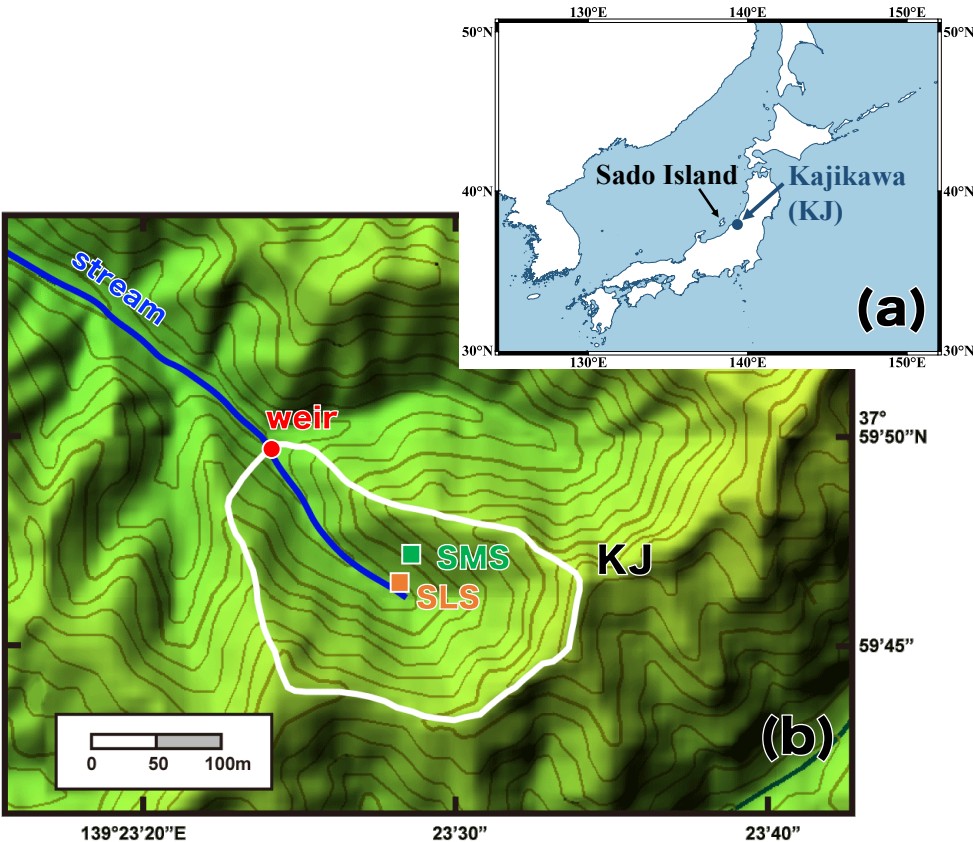

**Figure. 1** A map showing the locations of the studied Kajikawa (KJ) catchment in Japan
(a) and a colored altitude map of the KJ catchment (b) (modified after Nakagawa et al.
2018). The white line denotes the whole catchment area, and the red circle denotes the
position of the weir where the stream water was sampled. The orange (SLS) and green
(SMS) squares denote the sampling stations of soil water in the riparian and upland
zone, respectively, in the past study (Nakagawa et al., 2018).

2.2 Discharge rates and weather information
A V-notch weir (half angle: 30°) and a partial flume were installed at the bottom of
the catchment (Fig. 1b), where the discharge rates were determined. The weather
information including the precipitation monitored by Japan Meteorological Agency at
the nearest station of KJ catchment (Nakajo station; 38°04'60" N, 139°23'30" E) was
used for that in the KJ catchment. Because the accumulated snow was not monitored
in Nakajo station, however, those monitored at the Niigata station (37°53'60" N,
139°01'10" E) was used instead.

2.3 Sampling
In this study, the concentrations and stable isotopic compositions ($\delta^{15}N$, $\delta^{18}O$, and
$\Delta^{17}O$) of stream nitrate eluted from the KJ catchment were monitored every month for
more than 2 years (routine observation). Additionally, during storm events, the same
parameters were monitored every hour for 1 day (intensive observation). Stream water
was sampled at the weir located on the outlet of the KJ catchment (Fig. 1b). Routine
observation was performed manually using bottles at the weir approximately once a
month from 2017/5 to 2020/3. Intensive observation was conducted during the three
storm events I, II, and III (2019/8/22, 2019/10/12, and 2020/9/13, respectively), where
the water samples were collected at intervals of 1 hour over 24 hours using an automatic
water sampler (SIGMA 900, Hach, USA). In this study, 0.5 or 2 L polyethylene bottles
washed using chemical detergents were rinsed at least three times using deionized water
and dried in the laboratory before being used to store the water samples.

2.4 Analysis
Samples of stream water for the routine observation were transported to the
laboratory within 1 hour after being collected manually. Samples for the intensive
observation were transported within 12 days after completion of the automatic sampling
(Table 1). All samples were passed through a membrane filter (pore size 0.45 μm) and
stored in a refrigerator (4°C) until their chemical analysis.
The concentrations of nitrate were measured by ion chromatography (DX-500;
Dionex Inc., USA). To determine the stable isotopic compositions of nitrate in the
stream water samples, nitrate in each sample was chemically converted to $N_2O$ using a
method originally developed to determine the $^{15}N/^{14}N$ and $^{18}O/^{16}O$ ratios of seawater
and freshwater nitrate (McIlvin and Altabet, 2005) that was later modified (Konno et
al., 2010; Tsunogai et al., 2011; Yamazaki et al., 2011). In brief, 11 mL of each sample
solution was pipetted into a vial with a septum cap. Then, 0.5 g of spongy cadmium
was added, followed by 150 μL of a 1 M $NaHCO_3$ solution. The sample was then shaken
for 18-24 h at a rate of 2 cycles $s^{-1}$. Then, the sample solution (10 mL) was decanted
into a different vial with a septum cap. After purging the solution using high-purity
helium, 0.4 mL of an azide–acetic acid buffer, which had also been purged using high-
purity helium, was added. After 45 min, the solution was alkalinized by adding 0.2 mL
of 6 M NaOH.
Then, the stable isotopic compositions ($\delta^{15}N$, $\delta^{18}O$, and $\Delta^{17}O$) of the $N_2O$ in each vial
were determined using the continuous-flow isotope ratio mass spectrometry (CF-IRMS)
system at Nagoya University. The analytical procedures performed using the CF-IRMS
system were the same as those detailed in previous studies (Hirota et al., 2010; Komatsu
et al., 2008). The obtained values of $\delta^{15}N$, $\delta^{18}O$, and $\Delta^{17}O$ for the $N_2O$ derived from the
nitrate in each sample were compared with those derived from our local laboratory
nitrate standards to calibrate the values of the sample nitrate to an international scale
and to correct for both isotope fractionation during the chemical conversion to $N_2O$ and
the progress of oxygen isotope exchange between the nitrate derived reaction
intermediate and water (ca. 20 %). The local laboratory nitrate standards used for the
calibration had been calibrated using the internationally distributed isotope reference
materials (USGS-34 and USGS-35). In this study, we adopted the internal standard
method (Nakagawa et al., 2013, 2018; Tsunogai et al., 2014) to calibrate the stable
isotopic compositions of sample nitrate. In order to calibrate the differences in $\delta^{18}O$ of
$H_2O$ between the samples and those our local laboratory nitrate standards were added
for calibration, the $\delta^{18}O$ values of $H_2O$ in the samples were analyzed as well (Tsunogai
et al., 2010, 2011, 2014).
To determine whether the conversion rate from nitrate to $N_2O$ was sufficient, the
concentration of nitrate in the samples was determined each time we analyzed the
isotopic composition using CF-IRMS based on the $N_2O^+$ or $O_2^+$ outputs. We adopted
the $\delta^{15}N$, $\delta^{18}O$, and $\Delta^{17}O$ values only when the concentration measured via CF-IRMS
correlated with the concentration measured via ion chromatography prior to isotope
analysis within a difference of 10 %.
Three kinds of the local laboratory nitrate standards were used to determine the
isotopic compositions of stream nitrate, which had been named to be GG01 ($\delta^{15}N = -$
3.07 ‰, $\delta^{18}O = +1.10$ ‰, and $\Delta^{17}O = 0$ ‰), HDLW02 ($\delta^{15}N = +16.11$ ‰, $\delta^{18}O = +22.$
20 ‰), and NF ($\delta^{18}O = +54.14$ ‰, $\Delta^{17}O = +19.16$ ‰). Both GG01 and HDLW02 were
used to determine $\delta^{15}N$ and $\delta^{18}O$ of stream nitrate, and both GG01 and NF were used
to determine $\Delta^{17}O$ of stream nitrate. The standard errors of the mean in the isotopic
compositions ($\delta^{15}N$, $\delta^{18}O$, and $\Delta^{17}O$) determined through repeated measurements on
GG01 ($n = 3$), were $\pm0.17$ ‰ for $\delta^{15}N$, $\pm0.25$ ‰ for $\delta^{18}O$, and $\pm0.10$ ‰ for $\Delta^{17}O$, during
the measurements in this study. We repeated the analysis of $\delta^{15}N$, $\delta^{18}O$, and $\Delta^{17}O$ values
for each sample at least three times to attain high precision. All samples had a nitrate
concentration of greater than 10 μM, which corresponded to a nitrate quantity greater
than 100 nmol in a 10 mL sample. Thus, all isotope values presented in this study have
an error (standard error of the mean) better than $\pm0.2$ ‰ for $\delta^{15}N$, $\pm0.3$ ‰ for $\delta^{18}O$, and
$\pm0.1$ ‰ for $\Delta^{17}O$.
Nitrite ($NO_2^-$) in the samples interferes with the final $N_2O$ produced from nitrate
because the chemical method also converts $NO_2^-$ to $N_2O$ (McIlvin and Altabet, 2005).
Therefore, it is sometimes necessary to remove $NO_2^-$ prior to converting nitrate to $N_2O$.

However, in this study, all the stream and soil water samples analyzed for stable isotopic composition had $NO_2^-$ concentrations lower than the detection limit (0.05 µM). Because the minimum nitrate concentration in the samples was 24.7 µM in this study, the ratios of $NO_2^-$ to nitrate in the samples must be less than 0.2 %. Thus, we skipped the processes for removing $NO_2^-$.

2.5 Calculating of the concentration of unprocessed $NO_3^-{}_{atm}$ in stream water

The $\Delta^{17}O$ data of nitrate in each sample can be used to estimate the concentration of $NO_3^-{}_{atm}$ ($[NO_3^-{}_{atm}]$) in the stream water samples by applying Eq. (2):

$$[NO_3^-{}_{atm}]/[NO_3^-] = \Delta^{17}O/\Delta^{17}O_{atm} \tag{2}$$

where $[NO_3^-{}_{atm}]$ and $[NO_3^-]$ denote the concentration of $NO_3^-{}_{atm}$ and nitrate (total) in each water sample, respectively, and $\Delta^{17}O_{atm}$ and $\Delta^{17}O$ denote the $\Delta^{17}O$ values of $NO_3^-{}_{atm}$ and nitrate (total) in the stream water sample, respectively. In this study, we used the average $\Delta^{17}O$ value of $NO_3^-{}_{atm}$ determined at the nearby Sado-Seki monitoring station during the observation from April 2009 to March 2012 ($\Delta^{17}O_{atm} = +26.3$ ‰; Tsunogai et al., 2016) for $\Delta^{17}O_{atm}$ in Eq. (2) to estimate $[NO_3^-{}_{atm}]$ in the stream. We allow for an error range in of 3 ‰ in $\Delta^{17}O_{atm}$, in which the factor changes in $\Delta^{17}O_{atm}$ from +26.3 ‰ caused by both areal and seasonal variation in the $\Delta^{17}O$ values of $NO_3^-{}_{atm}$ have been considered (Nakagawa et al., 2018; Tsunogai et al., 2016).

**Table 1.** Information on the samples taken during the intensive observation.

| Storm event | Start time | End time | Date of filtration | Maximum period of storage without filtration |
|---|---|---|---|---|
| I | 2019/8/22 16:00 | 2019/8/23 15:00 | 2019/8/29 | 7 days |
| II | 2019/10/12 15:00 | 2019/10/13 14:00 | 2019/10/23 | 11 days |
| III | 2020/9/13 11:00 | 2020/9/14 10:00 | 2020/9/25 | 12 days |


**3 Results**
3.1 Variation during the routine observation
During the routine observation, the concentrations of stream nitrate ranged from
35.7 µM to 129.3 µM with the flux-weighted average concentration of 55.6 µM (Fig.
2a), showing little temporal changes from that determined during the past observations
from 2013 to 2014 at the same catchment (58.4 µM; Nakagawa et al., 2018). The
variation range also agreed with the past observation done in the same catchment
(Kamisako et al., 2008), except for the extraordinarily large concentration (129.3 µM)
recorded on 2018/8/31, which exceeded the $2\sigma$ of the whole variation range of stream
nitrate of our routine observation (Fig. 2a). We will discuss the reason in section 4.2.
The stable isotopic compositions of stream nitrate during the routine observation
ranged from +0.1 ‰ to +5.9‰ for $\delta^{15}N$ (Fig. 2b), from −1.9 ‰ to +7.7 ‰ for $\delta^{18}O$ (Fig.
2c), and from +0.4 ‰ to +2.7 ‰ for $\Delta^{17}O$ (Fig. 2d), while showing little seasonal
variation. The flux-weighted averages for the $\delta^{15}N$, $\delta^{18}O$, and $\Delta^{17}O$ values of nitrate
were +2.0 ‰, +1.1 ‰, and +1.1 ‰, respectively. Except for the extraordinarily large
$\delta^{18}O$ and $\Delta^{17}O$ values we found on 2019/1/31 ($\delta^{18}O$ = +7.7 ‰ and $\Delta^{17}O$ = +2.7 ‰)
(Figs. 2c and 2d), the values are typical for stream nitrate eluted from temperate forested
catchments (Hattori et al., 2019; Huang et al., 2020; Nakagawa et al., 2013, 2018; Riha
et al., 2014; Sabo et al., 2016; Tsunogai et al., 2014, 2016). On the other hand, the data
recorded on 2019/1/31 exceeded the $2\sigma$ variation range of the whole $\delta^{18}O$ and $\Delta^{17}O$
data. We will discuss the reason in section 4.3.

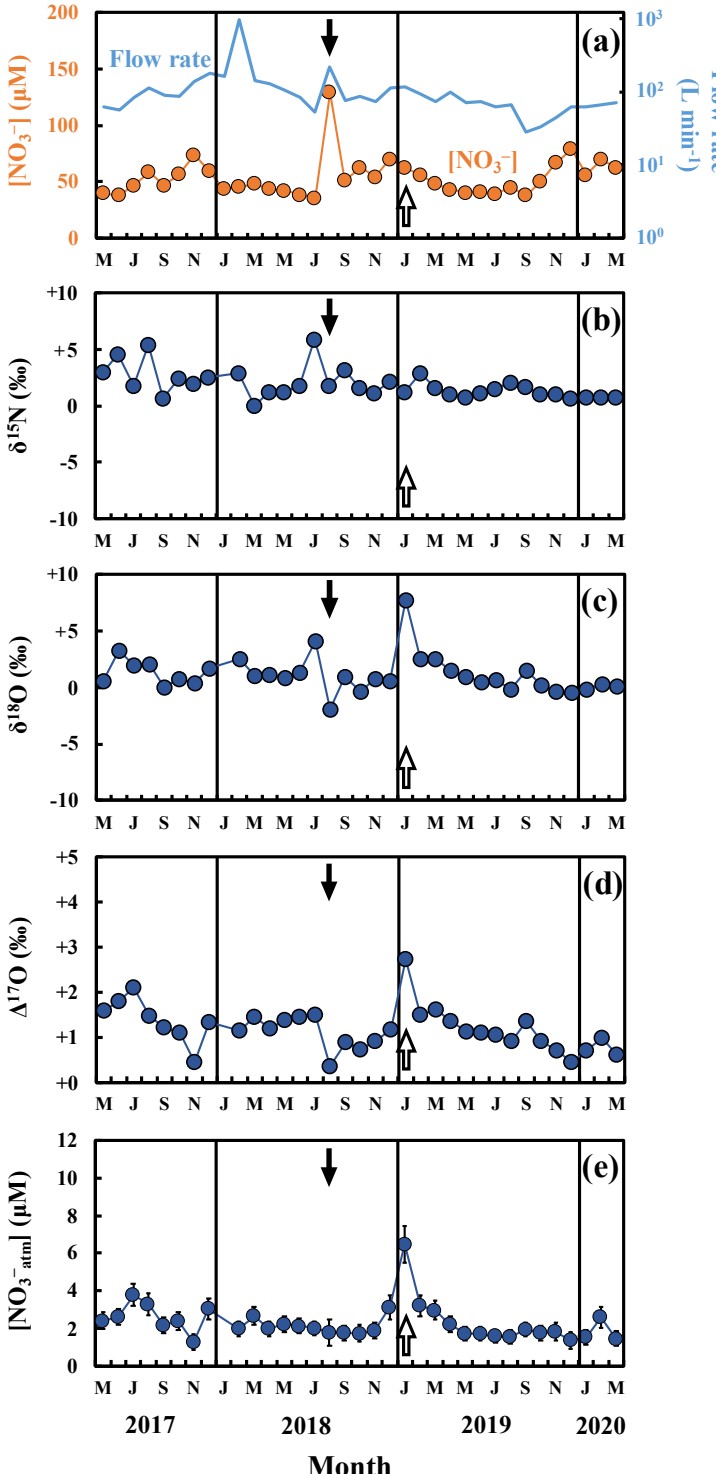

**Figure 2.** Temporal variations in the concentrations of nitrate (orange circles) and the

flow rates (blue line) in the stream water during the routine observation (a), together

with those of the values of $\delta^{15}N$ (b), $\delta^{18}O$ (c), $\Delta^{17}O$ (d) of nitrate, and the concentrations

of unprocessed atmospheric nitrate ($[NO_3^-{}_{atm}]$) (e) in the stream water (blue circles).
The black and white arrows in the figures indicate the sampling that took place on
2018/8/31 and 2019/1/31, respectively. The error bars smaller than the sizes of the
symbols are not presented.

3.2 Variation in response to the storm events
During the intensive observations made in response to the storm events, the
concentration of stream nitrate showed significant short-term variation, from 24.7 µM
to 122.6 µM, from 28.7 µM to 134.1 µM, and from 46.6 µM to 114.5 µM during the
storm events I, II, and III, respectively, with the minimum recorded just before the
beginning of each storm event and the maximum recorded when the flow rate was close
to the maximum within each storm event (Figs. 3 and S1). Similar increase in the
concentrations of stream nitrate in accordance with the increase in the flow rate during
storm events have been reported in many past studies (e.g. Burns et al.,2019; Chen et
al., 2020; Kamisako et al., 2008; Christopher et al., 2008). Especially, Kamisako et al.
(2008), who monitored temporal changes in the concentration of stream nitrate in the
same KJ catchment from 2003 to 2005 and found 11 nitrate increase events in
accordance with the increase in the flow rate, reported the largest concentration of
stream nitrate during the events to be 120 µM. The pattern and range of the short-term
variation of the stream nitrate concentration during the three storm events were also
consistent with the past study (Kamisako et al., 2008).
The stable isotopic compositions of stream nitrate during the three storm events also
showed significant temporal variation, from −0.1 ‰ to +2.5 ‰, from −0.5 ‰ to +3.0 ‰,
and from −0.1 ‰ to +3.5 ‰ for $\delta^{15}N$ (Figs. 3b, S1b, and S1g), from −3.4 ‰ to +3.1 ‰,
from −2.5 ‰ to +2.9 ‰, and from −2.3 ‰ to +2.1 ‰ for $\delta^{18}O$ (Figs. 3c, S1c, and S1h),
and from +0.3 ‰ to +1.6 ‰, from +0.3 ‰ to +1.4 ‰, and from +0.5 ‰ to +1.2 ‰ for
$\Delta^{17}O$ (Figs. 3d, S1d, and S1i), with minimum values observed when the concentration
of stream nitrate was at maximum and maximum values observed when the
concentration of stream nitrate was at a minimum.

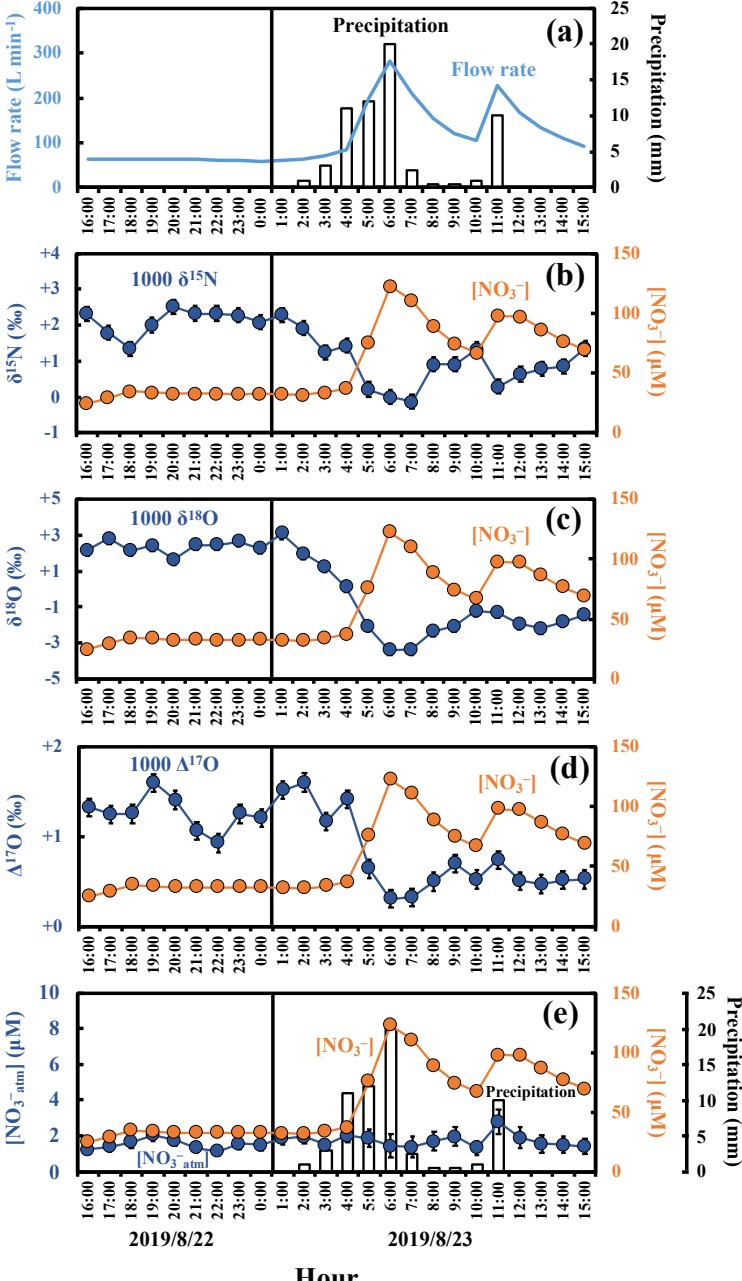

**Figure. 3** Temporal variations in the amount of precipitation (bar chart) and flow rates

of the stream water (blue line) during storm event I (a), together with those in the

concentrations of nitrate (orange circles) (b-e), the values of $\delta^{15}N$ (b), $\delta^{18}O$ (c), $\Delta^{17}O$ (d)

of nitrate, and [$NO_3^-{}_{atm}$] (e) in the stream water (blue circles). The error bars smaller

than the sizes of the symbols are not presented.

306

**4 Discussion**

4.1 Possible alterations to the concentration and isotopic compositions of stream nitrate

during the storage period in the automatic sampler used for the intensive observations

During the intensive observations, the stream water samples were stored in bottles of

the automatic sampler. The storage periods until filtration were ranged from 7 (storm

event I) to 12 days (storm event III) (Table 1). While the automatic sampler was

surrounded by ferns and the other understory vegetations to minimize the possible

alterations on the samples, progress of biogeochemical reactions such as nitrification,

denitrification, and assimilation could alter the concentration and isotopic compositions

($\delta^{15}N$, $\delta^{18}O$, and $\Delta^{17}O$) of stream nitrate during the storage period. Above all, possible

increase in soil water input into the stream water that is enriched with organic matters

during a storm event could enhance nitrification during the storage period and could

increase the concentration of nitrate in the stream water samples taken by using the

automatic sampler.

As a result, we discussed the possible alteration of the concentration and isotopic

compositions during the storage for the samples taken by using the automatic sampler

and concluded that the alterations during the storage in the automatic sampler were

minor in the samples. The details are described in Appendix A.

4.2 Primary source of nitrate increased during storm events

The striking feature of the observed short-term variation was that all the stable

isotopic compositions ($\delta^{15}$N, $\delta^{18}$O, and $\Delta^{17}$O) varied in response to the variation in the
nitrate concentration throughout the three storm events (Figs. 3 and S1). The result
implied that the source of increased nitrate during the storm events were different from
that during the base flow period.
As a result, the stable isotopic compositions ($\delta^{15}$N, $\delta^{18}$O, and $\Delta^{17}$O) of stream
nitrate were plotted as the functions of the reciprocal of the stream nitrate
concentration ($1/[NO_3^-]$) for each storm event (Fig. 4). All the stable isotopic
compositions of stream nitrate showed strong linear relationships ($R^2 > 0.5$; $p < 0.001$)
with the reciprocal of concentrations. The linear relationships strongly suggest mixing
between two endmembers with distinctively different isotopic signatures (e.g.
Keeling, 1958). The observed strong linear relationships not only in the $\Delta^{17}$O of
stream nitrate (Figs. 4g, 4h, and 4i), which is stable during the progress of partial
removal reactions such as denitrification or assimilation, but also in the $\delta^{15}$N and $\delta^{18}$O
of stream nitrate (Figs. 4a-4f), which should be altered during the progress of the
partial removal reactions, also implied that the progress of denitrification or
assimilation in bottles of the automatic sampler during the storage period without
filtration were minor in the samples.
The nitrate-depleted endmember must be the source of stream nitrate during the
base flow period prior to each storm event. On the other hand, the nitrate-enriched
endmember represents the source of nitrate that was added during the storm events.
Atmospheric nitrate ($NO_3^-{}_{atm}$) dissolved in rainwater was one of the possible
sources of nitrate enriched during the storm events (Inamdar and Mitchell, 2006).
While the $NO_3^-{}_{atm}$ showed the $\delta^{18}O$ and $\Delta^{17}O$ values enriched in both $^{18}O$ and $^{17}O$,
more than +55 ‰ and more than +18 ‰, respectively, during summer periods in
Japan (Tsunogai et al., 2016), the nitrate-enriched endmember showed the $\delta^{18}O$ and
$\Delta^{17}O$ values depleted in both $^{18}O$ and $^{17}O$, less than +3.1 ‰ and +1.6 ‰, respectively,
during the storm events. During storm events, increase in $\delta^{18}O$ and/or $\Delta^{17}O$ have been
reported for stream nitrate eluted from forested catchments in past studies (Sebestyen
et al., 2019; Sabo et al., 2016; Buda and Dewalle. 2009). In KJ catchment, however,
we found significant decrease in both the $\delta^{18}O$ and $\Delta^{17}O$ of stream nitrate during
storm events. In addition, the concentrations of $NO_3^-{}_{atm}$ ([$NO_3^-{}_{atm}$]) showed little
temporal variations showing the concentrations of $1.6 \pm 0.4$ µM, $1.8 \pm 0.4$ µM, and
$2.1 \pm 0.4$ µM during the storm events I, II, and III, respectively (Figs. 3e, S1e, and
S1j). In general, the [$NO_3^-{}_{atm}$] in rainwater were much higher than those in stream
water (Nakagawa et al., 2018; Rose et al., 2015; Tsunogai et al., 2014). During the
storm events I, II, and III, however, the [$NO_3^-{}_{atm}$] in stream water was almost constant
irrespective to the increase in precipitation (Figs. 3e, S1e, and S1j). Thus, we
conclude that the direct input of [$NO_3^-{}_{atm}$] via rainwater into the stream through
overland flow during storm events can be negligible, at least in the KJ catchment.
Thus, we concluded that the $NO_3^-{}_{atm}$ should be the minor source of nitrate that
increased during the storm events.
Nakagawa et al. (2018) determined the temporal variations in the concentrations
(Fig. 5a) and isotopic compositions ($\delta^{15}N$, $\delta^{18}O$, and $\Delta^{17}O$) (Figs. 5b, 5c, and 5d) of
soil nitrate dissolved in soil water taken within the same catchment during 2013 to
2014, at the depths of 20 cm and 60 cm of the station SLS (SLS 20 and SLS 60,
respectively) and at the depth of 20 cm of the station SMS (SMS 20), where the
station SLS was located in the riparian zone of the stream and the station SMS was
about 20 m away from the stream and located in the upland zone (Fig. 1b). The
concentrations of soil nitrate showed significant seasonal variation, with the higher
concentration in summer and the lower concentration in winter (Fig. 5a). Both the
$\delta^{18}O$ and $\Delta^{17}O$ values also showed significant seasonal variation, with the minimum
in summer and the maximum in winter (Figs. 5c and d). To verify if the soil nitrate is
the source of the stream nitrate that was added to the stream during the storm events,
we also plotted soil nitrate at each site (SLS 20, SLS 60 and SMS 20) of the same
season in Fig. 4. Because our intensive observations on the storm events were done in
summer (from August to October), the average concentration and the average isotopic
composition during summer (from August to October) were calculated (Table 2) and
plotted in Fig. 4. The error bars of each soil nitrate denote the standard deviation (SD)
of each isotopic composition (n =5 for each). We found that the isotopic compositions
($\delta^{15}N$, $\delta^{18}O$, and $\Delta^{17}O$) of soil nitrate in the riparian zone (SLS 20 and SLS 60; Table
2) were always plotted on the nitrate-enriched extension (lower $1/[NO_3^-]$ extension)
of the mixing line during the storm events I, II, and III (Fig. 4), while those of the soil
nitrate in the upland zone (SMS 20; Table 2) were somewhat deviated from the
nitrate-enriched extension of the mixing line, $\delta^{18}O$ especially (Figs. 4d, 4e, and 4f).
We conclude that the primary source of nitrate added during the storm events was the
soil nitrate in the riparian zone.
The "flushing hypothesis" has been proposed to explain the increase in stream
nitrate concentration in accordance with the increase in flow rate during storm events
(Creed et al., 1996; Hornberger et al., 1994). During the base flow periods, nitrate
accumulate in shallow, oxic soil layers due to the progress of nitrification. When
water level became higher during storm periods, concentration of stream nitrate
increased due to flushing of the soil nitrate accumulated in the shallow soil layers of
riparian zones into stream (Chen et al., 2020; Creed et al., 1996; Ocampo et al., 2006).
Our finding that the primary source of nitrate increased during the storm events was
the soil nitrate in the riparian zone is consistent with the "flushing hypothesis." We
conclude that the flushing of soil nitrate in the riparian zone into the stream due to
rising of both stream water and groundwater level was primarily responsible for the
increase in stream nitrate during the storm events (Fig. 6).
Within the whole dataset on the variation of the concentration of nitrate in the stream
determined by Kamisako et al. (2008), increases in the concentration of stream nitrate
to more than 20 μM in response to storm events were limited to the storm events that
occurred in the warm months, from June to November. As the concentrations of soil
nitrate in the riparian zone (SLS 20 and SLS 60) were much higher in the warm months
(734 μM ± 496 μM; from June to November) than in the cold months (156 ± 124 μM;
from December to May), such seasonal variation in the concentration of riparian soil
nitrate is consistent with the observed seasonality in the influence of storm events on
the stream nitrate concentration, where significant increase were limited to warm
months, whereas insignificant effects are observed during cold months.

415       The stream nitrate during storm events showed $\delta^{15}N$ values more depleted in $^{15}N$ than

those during the base flow periods (Figs. 3b, S1b, and S1g), probably due to the input
of riparian soil nitrate more depleted in $^{15}N$. Compared with the $\delta^{15}N$ values of stream
nitrate taken during the base flow periods of routine observations when precipitation
was less than 1 mm/day (Fig. 2b; Table S1), the riparian soil nitrate (SLS 20 and SLS
60; Table 2) showed the $\delta^{15}N$ values around 3.5 ‰ lower. The trend and the extent of
the $^{15}N$-depletion coincided well with those determined in the forested catchments in
past studies (Fang et al., 2015; Hattori et al., 2019). Fang et al. (2015), for instance,
reported significant differences between the $\delta^{15}N$ values of soil nitrate and those of
stream nitrate in six forested catchments in Japan and China, and proposed that the
kinetic fractionation due to the progress of denitrification during the elution of soil
nitrate into groundwater was responsible for the relative $^{15}N$-enrichment in stream
nitrate compared with soil nitrate. As a result, the observed temporal decrease in the
$\delta^{15}N$ value of stream nitrate during storm events also supported that the flushing of soil
nitrate showing $^{15}N$-depleted $\delta^{15}N$ values into the stream was responsible for the
elevated of nitrate concentrations during storm events.

431  As mentioned in section 3.1, we found significant increase in nitrate concentration

432 up to 129.3 μM on 2018/8/31 during our routine observation on the stream, when the

433 water was sampled in the middle of a heavy storm (48.0 mm/day; Table S1) with

434 significant increase in flow rate (from 53.4 L/min one month before to 216.9 L/min

435 during sampling), which the amount of precipitation on 2018/8/31 was the highest

436 within the whole routine observations (Table S1). The measured $\delta^{18}O$ and $\Delta^{17}O$ value

437 of the stream nitrate on 2018/8/31 (−1.9 ‰ and +0.4 ‰, respectively), showing

438 significantly smaller values than those during the other routine observation (Fig. 2c

439 and 2d), agreed well with those of the nitrate increase during the storm events I, II,

440 and III. Moreover, both the range of increase in stream nitrate concentration (129.3

441 μM) and the season of observation (August) also agreed well with those of the stream

442 nitrate increase during the three storm events. As a result, we conclude that the input

443 of soil nitrate accumulated in the riparian zone due to flushing was also responsible

444 for the significant increase in stream nitrate concentration we found on 2018/8/31

445 during the routine observation.

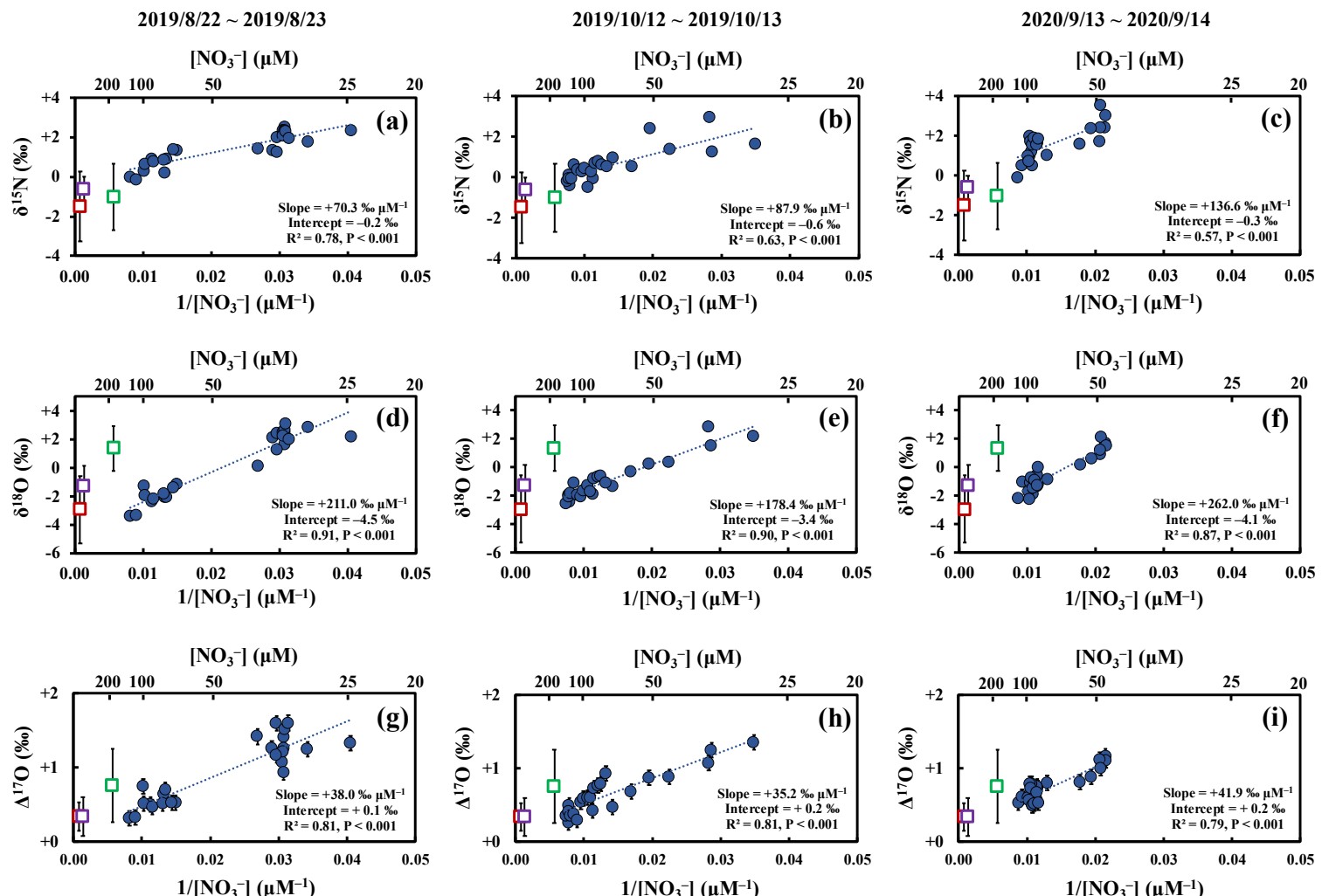

**Figure 4.** The $\delta^{15}N$ (a, b, and c), $\delta^{18}O$ (d, e, and f), and $\Delta^{17}O$ (g, h, and i) values of

stream nitrate (blue circles) during storm events I, II, and III plotted as a function of the

reciprocal of nitrate concentration (1/[$NO_3^-$]), together with those of soil nitrate at SLS

20 (red squares; riparian zone), SLS 60 (purple squares; riparian zone), and SMS 20

(green squares; upland zone) during August to October in 2013 and 2014. The error

bars of each soil nitrate denote the standard deviation (SD) of each isotopic composition

(n =5 for each). The error bars smaller than the sizes of the symbols are not presented.

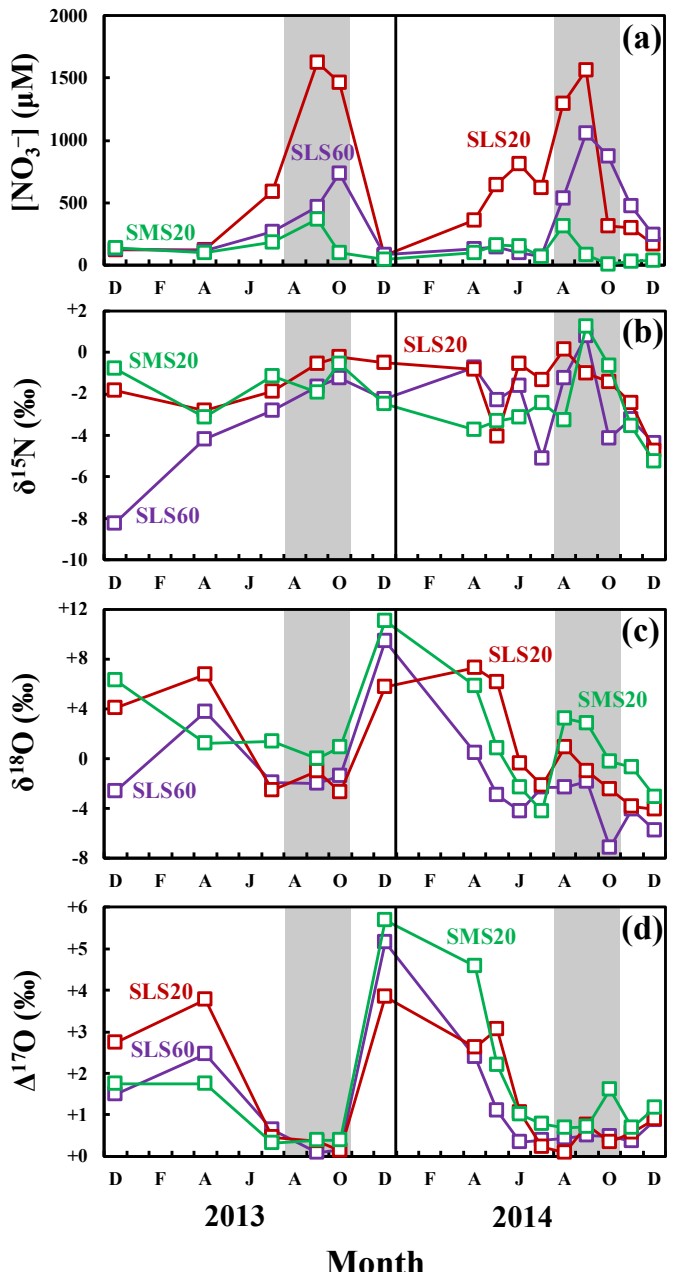

**Figure 5.** Seasonal variations in the concentrations of soil nitrate (a) at SLS 20 (red

squares), SLS 60 (purple squares), and SMS20 (green squares), together with those in

the values of $\delta^{15}N$ (b), $\delta^{18}O$ (c) and $\Delta^{17}O$ (d) of each soil nitrate during 2013 to 2014

(modified from Nakagawa et al., 2018). The periods used to estimate the isotopic

compositions (from August to October) are presented in gray. The error bars were

smaller than the sizes of the symbols.

**Table 2.** Concentrations and isotopic compositions ($\delta^{15}$N, $\delta^{18}$O, and $\Delta^{17}$O) of soil nitrate at SLS 20, SLS 60, and SMS 20 during August to October in 2013 and 2014 (recalculated from the data in Nakagawa et al., 2018).

|  | SLS 20 | SLS 60 | SMS 20 |
|---|---|---|---|
| $NO_3^-$ (μM) | $1254 \pm 537$ | $734 \pm 241$ | $176 \pm 159$ |
| $1000\ \delta^{15}$N | $-1.5 \pm 1.8$ | $-0.6 \pm 0.6$ | $-1.0 \pm 1.7$ |
| $1000\ \delta^{18}$O | $-2.9 \pm 2.4$ | $-1.3 \pm 1.4$ | $+1.4 \pm 1.6$ |
| $1000\ \Delta^{17}$O | $+0.3 \pm 0.2$ | $+0.3 \pm 0.3$ | $+0.8 \pm 0.5$ |

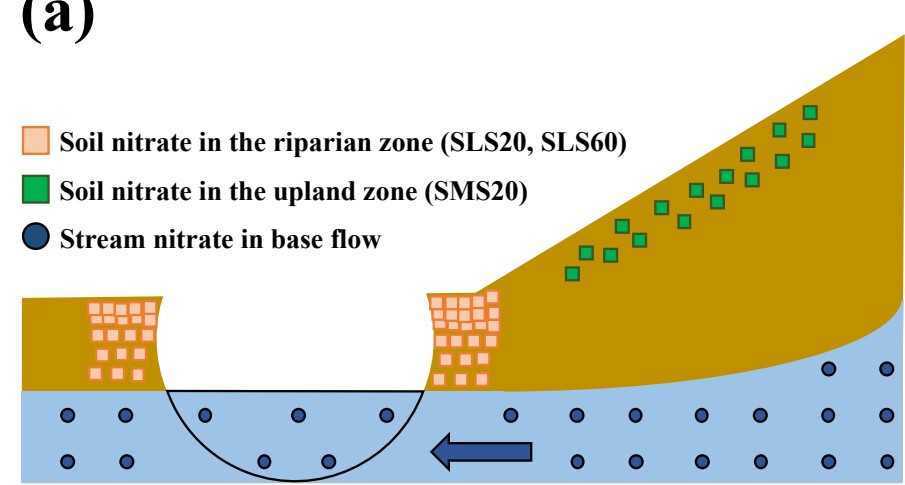

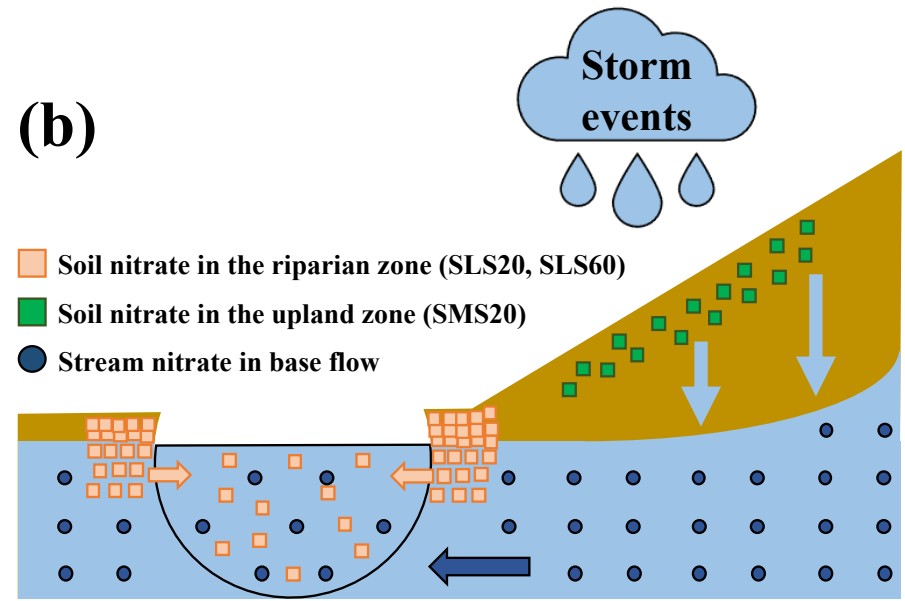

**Figure 6.** Schematic diagram showing the elution of soil nitrate to the stream before
the storm events (a) and during the storm events (b). Soil nitrate in the riparian zone
and that in the upland zone are represented by the orange squares and green squares,
respectively, while stream nitrate during base flow is represented by the blue circles.

4.3 Variation in the concentration of $NO_3^-{}_{atm}$ during routine observation
The concentration of $NO_3^-{}_{atm}$ ($[NO_3^-{}_{atm}]$) showed little seasonal variation, from 1.3
μM to 3.8 μM during our routine observation in this study (Fig. 2e), except for the
extraordinarily large $[NO_3^-{}_{atm}]$ we found on 2019/1/31 (6.5 μM). Except for the
extraordinarily large $[NO_3^-{}_{atm}]$, the obtained $[NO_3^-{}_{atm}]$ corresponded well with those
determined in the past study done at the same catchment (Nakagawa et al., 2018). In
addition, they corresponded well with those of the temperate forested catchments
saturated in nitrogen, such as Fernow experimental Forest 3 (4.2 μM; Rose et al., 2015).
In this study, accumulation of snow up to 18 cm was observed at the KJ catchment
on 2019/1/27, while most of the accumulated snow had melted to a depth of 1 cm by
2019/1/30, just before the sampling on 2019/1/31. Furthermore, during the routine
observation period from 2017/5 to 2020/3, no other snow-melting events occurred
within 4 days prior to the day of sampling, except for the sampling on 2019/1/31.
Similar enhancement in the concentration of $NO_3^-{}_{atm}$, as well as the $\delta^{18}O$ and $\Delta^{17}O$ of
stream nitrate, in response to snow melting has been frequently observed in streams
worldwide (Ohte et al., 2004, 2010; Pellerin et al., 2012; Piatek et al., 2005; Rose et al.,
2015; Sabo et al., 2016; Tsunogai et al., 2014, 2016).

489       The flow rate, concentration of stream nitrate, and $\Delta^{17}O$ was 110.0 L/min, 70.0 µM,

and +1.17 ‰ on 2018/12/28, respectively, while 117.3 L/min, 62.4 µM, and +2.73 ‰
on 2019/1/31, respectively (Table S1). The $[NO_3{}^-{}_{atm}]$ in stream water was estimated to
be 3.1 µM on 2018/12/28 and 6.5 µM on 2019/1/31. Assuming that the $[NO_3{}^-{}_{atm}]$ in
snow melt was the same with the volume-weighted mean concentration of nitrate in
rainwater (41.0 µM) determined at Sado island in January (EANET, 2010, 2011;
Tsunogai et al., 2016), the increase in the flow rate ($\Delta F_{snowmelt}$) due to the mixing of
snow melt into the stream can be estimated to be 10.3 L/min, by using the mass balance
equation shown below:
$([NO_3{}^-{}_{atm}]_{2019/1/31} \times F_{2019/1/31} = [NO_3{}^-{}_{atm}]_{2018/12/28} \times F_{2018/12/28} + [NO_3{}^-{}_{atm}]_{snowmelt} \times$
$\Delta F_{snowmelt})$                                                             (3)
where $[NO_3{}^-{}_{atm}]_{2018/12/28}$, $[NO_3{}^-{}_{atm}]_{2019/1/31}$, and $[NO_3{}^-{}_{atm}]_{snowmelt}$ denote the $[NO_3{}^-{}_{atm}]$ in
stream water on 2018/12/28, 2019/1/31, and that in snow melt water, respectively, and
$F_{2018/12/28}$, $F_{2019/1/31}$, and $\Delta F_{snowmelt}$ denote the flow rate of stream water on 2018/12/28,
2019/1/31, and the increase in the flow rate due to snow melt, respectively. Because the
estimated volume of melting snow water into the stream water (10.3 L/min) was
comparable with the observed increase in the flow rate from 2018/12/28 to 2019/1/31
(7.3 L/min), we concluded that the snow melting was responsible for the increase in
$\Delta^{17}O$ on 2019/1/31 and that the input of $NO_3{}^-{}_{atm}$ accumulated in the melted snow water,
showing $\delta^{18}O$ and $\Delta^{17}O$ values significantly higher than those in the stream, caused the

extraordinarily increase in $[NO_3^-{}_{atm}]$ on 2019/1/31. Except for the extraordinarily

increase in $[NO_3^-{}_{atm}]$ (n = 1), $[NO_3^-{}_{atm}]$ was stable at $2.2 \pm 0.6$ μM throughout the

routine observation (n = 33). We concluded that $[NO_3^-{}_{atm}]$ was generally stable in the

stream.

4.4 The impact of storm events on the index of the nitrogen saturation

 The concentration of stream nitrate eluted from a forested catchment has been used

as an index to evaluate the stage of nitrogen saturation (Huang et al., 2020; Rose et al.,

2015; Stoddard, 1994). However, McHale et al. (2002) pointed out the problem in the

reliability of this index, because the number of storm events influenced the

concentration of nitrate eluted from forested stream significantly. That is, if we use the

concentration of stream nitrate sampled during the storm events to evaluate the stage of

nitrogen saturation in a forested catchment, the stage of nitrogen saturation might be

overestimated.

 Nakagawa et al. (2018) have proposed the export flux of $NO_3^-{}_{atm}$ ($M_{atm}$) relative to

the deposition flux of $NO_3^-{}_{atm}$ ($D_{atm}$) can be an alternative, more robust index for

nitrogen saturation in temperate forested catchments, because the $M_{atm}/D_{atm}$ ratio

directly reflect the demand on atmospheric nitrate deposited onto each forested

catchments as a whole, and thus reflect the nitrogen saturation in each forested

catchment. To estimate reliable $M_{atm}$ in each forested catchment, we must obtain

reliable $[NO_3^-{}_{atm}]$ in the forested stream, including their temporal variation.

As already presented in section 4.2, we found that $[NO_3^-{}_{atm}]$ remained almost
constant irrespective to the significant variation in $[NO_3^-]$ during storm events (Figs.
3e, S1e, and S1j). The concentrations of atmospheric nitrate ($[NO_3^-{}_{atm}]$) in rainwater
were much higher than those in stream water. While the volume-weighted mean
$[NO_3^-{}_{atm}]$ in rainwater determined in Sado island from August to October, for example,
was $15.2 \pm 8.4$ μM (EANET, 2010, 2011; Tsunogai et al., 2016), that in the stream water
was $2.2 \pm 0.6$ μM in this study. As a result, the $[NO_3^-{}_{atm}]$ in stream water would increase,
if significant portion of rainwater was added directly into the stream water during the
storm events. The $[NO_3^-{}_{atm}]$ in stream water, however, was stable showing no
correlation with the amount of precipitation or the concentration of stream nitrate during
the storm events (Figs. 3e, S1e, and S1J). The $[NO_3^-{}_{atm}]$ remained almost constant as
well during the stream event on 2018/8/31 we found through the routine observation,
while $[NO_3^-]$ increased from 35.7 μM (1 month before) to 129.3 μM (Fig. 2e). As a
result, we concluded that the direct input of $NO_3^-{}_{atm}$ into the stream water was negligible
even during the storm events.
The observed $[NO_3^-{}_{atm}]$ showing almost constant values implies that the primary
source of $NO_3^-{}_{atm}$ in stream water during storm events was the $NO_3^-{}_{atm}$ stored in
groundwater, which is the same source as that during the base flow periods, rather than
the direct input of $NO_3^-{}_{atm}$ from rainwater. Because direct input of $NO_3^-{}_{atm}$ into stream
water was negligible during the storm events, the $M_{atm}/D_{atm}$ ratio in each forested
catchment should be controlled by the metabolized processes (uptake or denitrification)
in each forested catchment subsequent to deposition, so that the $M_{atm}/D_{atm}$ can correctly
reflect the total demand on $NO_3^-{}_{atm}$ in each forested catchment and thus the status of
nitrogen saturation. We conclude that the $M_{atm}/D_{atm}$ ration can be a more robust index
to evaluate nitrogen saturation in forested catchments.

**5 Conclusions**
Temporal variations in the concentrations and stable isotopic compositions ($\delta^{15}N$,
$\delta^{18}O$, and $\Delta^{17}O$) of stream nitrate were determined during storm events to clarify the
source of stream nitrate increased during storm events. Because the stable isotopic
compositions of soil nitrate in riparian zone during summer agreed well with those of
the nitrate-enrich endmember of the stream nitrate increased during storm events, we
conclude that the soil nitrate in riparian zone was primarily responsible for the stream
nitrate increase during storm events. Additionally, the concentration of $NO_3^-{}_{atm}$ in the
stream was almost constant during the storm events, implied that the source of $NO_3^-{}_{atm}$
in stream water during storm events was the $NO_3^-{}_{atm}$ stored in groundwater. We
concluded that the number of storm events have little impact on $M_{atm}/D_{atm}$ ratio, the
index of nitrogen saturation. In addition, the $\Delta^{17}O$ of nitrate can be applicable as the
tracer to clarify the source of nitrate.



**Appendix A: Possible alterations to the concentration and isotopic compositions of stream nitrate during the storage period in the automatic sampler used for the intensive observations**

During the intensive observations, the stream water samples were stored in bottles of the automatic sampler. The storage periods until filtration were ranged from 7 (storm event I) to 12 days (storm event III) (Table 1). While the automatic sampler was surrounded by ferns and the other understory vegetations to minimize the possible alterations on the samples, progress of biogeochemical reactions such as nitrification, denitrification, and assimilation could alter the concentration and isotopic compositions ($\delta^{15}$N, $\delta^{18}$O, and $\Delta^{17}$O) of stream nitrate during the storage period. Above all, possible increase in soil water input into the stream water that is enriched with organic matters during a storm event could enhance nitrification during the storage period and could increase the concentration of nitrate in the stream water samples taken by using the automatic sampler. Here, we discussed the possible alteration of the concentration and isotopic compositions during the storage for the samples taken by using the automatic sampler.

First, we compared the samples taken during the intensive observations using the automatic sampler with those taken during the routine observations. During the routine observations, the stream water samples were taken manually, transported to the laboratory within 1 h of each collection, passed through a membrane filter (pore size 0.45 μm), and stored in a refrigerator (4°C) until chemical analysis. As a result,

alterations should be minor in the samples taken through the routine observations.
When we compared the concentrations and isotopic compositions of stream nitrate
in the samples taken at the beginning of the intensive observation using the automatic
sampler with those in the routine observation nearby, they coincided well each other
(Table A1), implying that at least the progress of nitrification within the bottles of the
automatic sampler should be minor during the storage period because the concentration
of nitrate should increase, while the $\Delta^{17}O$ should decreased significantly during the
storage period if the progress of nitrification was active in the bottles of the intensive
observation.
In addition, a clear storm event was also observed during the routine observation on
2018/8/31 (Fig. 2; Table S1), so that we can compare the concentrations and isotopic
compositions of stream nitrate with those of intensive observations. During the routine
observation on 2018/8/31 done under a precipitation and flow rate of 48 mm/day and
216.9 L/min, respectively, we observed a significant increase in the concentration of
stream nitrate from 35.7 µM one month before to 129.3 µM (Fig. 2 and Table S1). In
accordance with the increase in the concentration, we found significant changes in the
isotopic compositions; from +5.9 ‰ to +1.8 ‰ for $\delta^{15}N$, from +4.1 ‰ to –1.9 ‰ for
$\delta^{18}O$, from +1.5 ‰ to +0.4 ‰ for $\Delta^{17}O$ (Fig. 2 and Table S1). The trend and the degree
of the variations in the concentration and the isotopic compositions on 2018/8/31 from
those on one month before were consistent with those of the intensive observation (Figs.
3 and S1). As a result, we concluded that the increase in the flow rate was responsible
for the observed increase in concentrations of stream nitrate during the storm events
and thus the microbial production of nitrate through nitrification during the storage had
little influence on the observed temporal changes in the concentrations and isotopic
compositions of nitrate in the stream water samples taken by using the automatic
sampler.
Kotlash and Chessman (1998) conducted storage experiments under various
conditions such as freezing, acidification, refrigeration, and room temperature to clarify
the changes in the concentrations of nitrogen compounds in stream water samples and
found little change in concentration of oxidized nitrogen ($NO_3^- + NO_2^-$) irrespective of
the treatments. To further verify the insignificant changes in the concentrations and
isotopic compositions of stream nitrate stored without treatments in the samples taken
by the automatic sampler, we also conducted the storage experiments by using a 100
mL of stream water taken at the KJ forested catchment on 2022/4/28 and stored in a
100 ml PP (polypropylene) bottle without treatments. Approximately 85 mL of the
stream water within the bottle was filtered using a GF/F filter paper (25 mm diameter)
and stored in a refrigerator (4°C) to determine original (initial) concentration and
isotopic compositions of nitrate. To simulate the stream water containing increased
suspended organic matters during the storm events, the GF/F filter paper was returned
to the 100 mL PP bottle which contained 15 mL of the stream water sample and left the
15 mL stream water under the room temperature (18.3°C) for 2 weeks together with the
suspended organic matters on the filter. The concentration and isotopic compositions of
the original stream nitrate (85 mL) and those being stored without filtration under the
room temperature for 2 weeks (15 mL) were analyzed by using the same method
presented in section 2.4. The concentration of nitrate in the stream water sample being
stored for 2 weeks without treatments coincided well with those in the original, showing
the difference in concentrations less than 10 % (Table A2). Besides, the differences in
the isotopic compositions from the original were also negligibly small (Table A2).
As a result, we concluded that the possible alteration in the concentration and isotopic
compositions of nitrate due to the progress of biogeochemical reactions such as
nitrification, denitrification, and assimilation during storage in the automatic sampler
used in the intensive observations was negligibly small.











**Table A1.** Comparison of both concentration and isotopic compositions ($\delta^{15}N$, $\delta^{18}O$,
and $\Delta^{17}O$) of stream nitrate between those taken at the beginning of intensive
observations using the automatic sampler and those taken manually on the days nearby
during routine observations.

| | Type | Flow rate L/min | Precipitation mm/day | $NO_3^-$ μM | $\delta^{15}N$ /$10^3$ | $\delta^{18}O$ /$10^3$ | $\Delta^{17}O$ /$10^3$ |
|---|---|---|---|---|---|---|---|
| 2019/7/31 | routine | 61.6 | 0.0 | 39.5 | +1.55 | +0.66 | +1.06 |
| 2019/8/22 16:00 | intensive | 64.1 | 1.0 | 24.7 | +2.32 | +2.17 | +1.33 |
| 2019/8/30 | routine | 66.0 | 13.0 | 44.9 | +2.07 | −0.13 | +0.91 |
| | | | | | | | |
| 2019/9/30 | routine | 28.0 | 0.0 | 37.9 | +1.65 | +1.56 | +1.36 |
| 2019/10/12 15:00 | intensive | 22.4 | 7.0 | 28.7 | +1.61 | +2.18 | +1.35 |
| 2019/10/31 | routine | 32.6 | 0.0 | 50.4 | +1.04 | +0.19 | +0.92 |
| | | | | | | | |
| 2020/9/13 11:00 | intensive | 111.0 | 0.0 | 46.6 | +2.42 | +1.74 | +1.17 |
| 2020/9/30 | routine | 117.3 | 0.0 | 63.2 | − | − | − |

−:No samples were taken for isotopic analysis

Table A2. Comparison of both concentration and isotopic compositions ($\delta^{15}N$, $\delta^{18}O$,
and $\Delta^{17}O$) between original stream water sample and that being stored under the room
temperature for 2 weeks without treatments.

| | $NO_3^-$ μM | $\delta^{15}N$ /$10^3$ | $\delta^{18}O$ /$10^3$ | $\Delta^{17}O$ /$10^3$ |
|---|---|---|---|---|
| Original | 53.2 | +0.90 | +0.80 | +1.05 |
| Stored | 49.5 | +0.85 | +0.90 | +0.99 |


*Data availability.* All the primary data are presented in the Supplement. The other data
are available upon request to the corresponding author (Weitian Ding).

*Author contributions.* WD, UT, NY, and HS designed the study. HY, MM, and HS performed the field observations. HY, MM, and HS determined the concentrations of the samples. WD determined the isotopic compositions of the samples. WD, TS, FN, and UT performed data analysis, and WD and UT wrote the paper with input from MM, HY and HS.

*Competing interests.* The authors declare that they have no conflict of interest.

*Acknowledgements.*

We thank anonymous referees for valuable remarks on an earlier version of this paper. The samples analyzed in this study were collected through the Long-term Monitoring of Transboundary Air Pollution and Acid Deposition by the Ministry of the Environment in Japan. The authors are grateful to Ryo Shingubara, Masanori Ito, Hao Xu, Hui Lan, Peng Lai, Tianzheng Huang, Yuhei Morishita, Tae Ito, Yuka Tadachi and other present and past members of the Biogeochemistry Group, Nagoya University, for their valuable support throughout this study. This work was supported by a Grant-in-Aid for Scientific Research from the Ministry of Education, Culture, Sports, Science, Technology of Japan under grant numbers JP17H00780, JP19H04254, and JP19H00955, the Yanmar Environmental Sustainability Support Association, and the River Fund of The River Foundation, Japan. Weitian Ding would

like to take this opportunity to thank the 'Nagoya University Interdisciplinary Frontier
Fellowship' supported by JST and Nagoya University.

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
