# Peer review of "Tracing the source of nitrate in a forested stream showing elevated concentrations during storm events"

_Biogeosciences, 2022_

## Author Comment (AC1)

Reply to your comment (Referee #1).

Thank you very much for your valuable comments on our manuscript. We would like to respond to each of your comments and questions one by one.

> For example, Sebestyen et al. 2019 (ES&T) and the references it contains address a similar issue as this manuscript (e.g. Buda et al. 2009 and Sabo et al. 2016 both sampled storm events), and this manuscript could do a more throughout job of using those studies to help justify this study (in the Introduction) and then comparing/contrasting the results of this study to those studies in the Discussion. Similarly, oher studies (Burns et al. 2009; Barnes et al. 2010; Bostic et al. 2021) have addressed similar questions in non-forested systems and could be useful for helping to provide a broader context for the results that are presented in this manuscript.

Thank you for your advising. We would like to cite many of the suggested articles in the revised manuscript.

> Title: "Enriched" is a word that is often used incorrectly in the isotope literature to refer to increased values of the heavier isotope. Here I believe the authors use "enriched" to mean increased nitrate concentrations, which is might cause confusion given that this paper also talks about isotopic enrichment (e.g lines 66 and 303). One solution might be to simply delete "enriched" from the title and another solution might be replace "nitrate enriched" in the title with something like "elevated nitrate concentrations"

Thank you for your advising. We would like to revise the title as "Tracing the source of nitrate in a forested stream showing elevated concentrations during storm events" in the revised manuscript.

> Lines 2-3: This sentence implies that nitrate concentrations always increase in temperate forest streams everywhere. Is that true? If not, perhaps slightly adjust this sentence.

While many past studies reported increasing the stream nitrate concentration during storm events in temperate forest (e.g. Creed et al., 1996; Kamisako et al., 2008; Christopher., 2008), the decrease pattern (Christopher., 2008) or stable pattern (Shanley et al., 2011) of stream nitrate concentration during storm events also have been reported. Thank you for your advising. We would like to revise the sentence as suggested.

> For example, do some severely nitrogen saturated forests that show higher NO3 concentrations in baseflow than stormflow?

No. For example, the KJ forested catchment has been reported under severely nitrogen saturation (Nakagawa et al., 2018), both this study and past study (Kamisako et al., 2008) found significant increase in the stream nitrate concentration during stormflow than baseflow. Thank you for your advising.

> Line 5: Please tell the reader what time of year (winter, spring, summer, autumn) these storm events occurred.

Thank you for your advising. We would like to revise as suggested.

> Line 6: It might be helpful to insert "increasing" before "from" to help the reader understand that the "variation" nitrate concentration that was observed was primarily an increase in concentrations.

Thank you for your advising. We would like to revise as suggested.

> Line 14: I believe "($d^{15}N$, $d^{18}O$, and $C^{17}O$)" can be deleted without sacrificing meaning.

Thank you for your advising. We would like to revise as suggested.

> Line 26-27: Could the authors support this claim by calculating annual export of NO3-atm (and NO3-terr) using their concentration and flow data?

The annual export of unprocessed atmospheric nitrate in the stream can be calculated as $3.2 \pm 0.7$ mmol m$^{-2}$ yr$^{-1}$ by multiplying the average flow rate of stream and the average concentration of unprocessed atmospheric nitrate in the stream during the routine observation.
The annual export flux of unprocessed atmospheric nitrate relative to the annual deposition flux ($M_{atm}/D_{atm}$ ratio), nitrogen saturation index, was estimated from annual concentration of unprocessed atmospheric nitrate in the stream, annual flow rate of stream, and annual deposition flux of atmospheric nitrate. In the forested catchment, the annual flow rate of stream and annual deposition flux of atmospheric nitrate can be considered as constant. The concentration of unprocessed atmospheric nitrate in the stream was $1.6 \pm 0.4$ μM, $1.8 \pm 0.4$ μM, and $2.1 \pm 0.4$ μM during the storm events I, II, and III, respectively, which have no significant difference with the average concentration of unprocessed atmospheric nitrate in the stream ($2.2 \pm 0.6$ μM). Thus, the storm events have little impacts on the $M_{atm}/D_{atm}$ ratio. Thank you for your advising. We would like to clarify this in the revised MS.

> Lines 26-30: Is this conclusion specific to the author's study site (or certain types of forests) or are they suggesting that is a more broad/general conclusion that applies to forested catchments everywhere?

We conducted the research at KJ forested catchments as an example of the nitrogen saturated forested catchment. The conclusion is only suitable for the KJ forested catchment at present. Further works should be needed to verify the conclusion in different forested catchments in the future.

> Line 33: "representative" of what?   Please clarify.

Thank you for your advising. We will clarify this in the revised manuscript.

> Line 50: First, how are the authors using "overland flow" here and elsewhere (e.g. line 463) in the manuscript?   My understanding is that overland flow is unlikely in areas that are not near channels or stream/riparian areas in forests except for unique situations, such as intense rain events or rain that occurs on frozen soils. Second, I don't believe either of the cited studies suggest that overland flow is a mechanism for direct suppler of atmospheric nitrate to stream water. As far as I recall, Kaushal et al. didn't show overland flow for their forested site and Sebestyen et al. talked about routing of NO3-atm along flow paths that allowed NO3-atm to bypass uptake/processing (but not specifically about overland flow).

Thank you for your advising. We would like to revise this in the revised manuscript.

> Line 72: Is beta completely constant or can it exhibit some variation around 0.5279? If so, does the variation affect the authors data analyses or interpretations?

Bostic et al. (2021) assumed the $\beta$ could vary from 0.51 to 0.53. In the whole samples analyzed in this study (n=105), both the min and max value of the $\delta^{18}O$ was -3.3 ‰ and +7.7 ‰, respectively. Thus, the max range of deviation in the $\Delta^{17}O$ could be estimated to be 0.1 ‰ (Fig 1), which in accordance with our analytical standard error of $\Delta^{17}O$. As a result, our conclusion cannot be influenced by the variation of $\beta$. Thank you for your advising.

[Figure]

Figure 1. Schematic diagram showing the max variation of the $\Delta^{17}O$ in accordance with the variation of the $\beta$ from 0.51 to 0.53 in this study.

> Lines 162-164: It seems like there would be potential for microbial alteration of the samples during the 1-2 weeks that they stayed in the field before being returned to the lab. Did the authors assess this?

We think the microbial alteration of the stream water samples during the storage period can be negligible.
(1) The sampler was set on the riverbank near a weir surrounded by ferns and other understory vegetation avoiding sunlight during the observation. In addition, the bottles are stored in a shaded space to minimize the microbial alteration of the samples. Besides, the automatic sampler (SIGMA 900, Hach, USA) has equipped with refrigerator to keep the samples in 4°C. (2) Kotlash and Chessman (1998) have assessed the storage effects of freezing, acidification, refrigeration and extended storage without refrigeration (6 days) on measured concentrations of nitrogen of different stream water samples, and found there was little difference in concentration of oxidized nitrogen ($NO_3^-$ + $NO_2^-$) according to different treatment. (3) The concentrations of stream nitrate showed temporal variation in accordance with the variation in the stream flow rate during storm events (Figs. 3 and S1). As a result, the variation of the stream nitrate concentrations was primarily controlled by the flow rate instead of the microbial process. (4) The $\Delta^{17}O$ of stream nitrate is stable during the progress of such microbial processes (e.g., denitrification or assimilation). While the $\delta^{15}N$ and $\delta^{18}O$ of stream nitrate can be altered by the progress of partial removal through microbial process, the $\delta^{15}N$ and $\delta^{18}O$ of stream nitrate showed strong linear relationship between the reciprocal of concentrations, implying that the primary

process controlling both $\delta^{15}$N and $\delta^{18}$O was mixing. Thank you for your advising. We would like to emphasize this in the revised manuscript.

> Lines 184-185: How many "local laboratory nitrate standards" were used and what are their isotope values?
> Lines 205-206: What data were used to calculate the reported standard error of the mean for each isotope? For example, was precision determined from the lab standards, replicate samples, or something else?

In this study, we used three kinds of the local laboratory nitrate standards, which were named to be GG01 ($\delta^{15}$N = –3.07 ‰, $\delta^{18}$O = +1.10 ‰, and $\Delta^{17}$O = 0 ‰), HDLW02 ($\delta^{15}$N = +16.11 ‰, $\delta^{18}$O = +22.20 ‰), and NF ($\Delta^{17}$O = +19.16 ‰), which the GG01 and the HDLW02 were used to determine the $\delta^{15}$N and $\delta^{18}$O of stream nitrate, and the GG01 and the NF was used to determine the $\Delta^{17}$O of stream nitrate. The standard error of the mean of the isotopic compositions ($\delta^{15}$N, $\delta^{18}$O, and $\Delta^{17}$O) was determined by repeated measurements of the GG01 (n = 3), ±0.17 ‰ for $\delta^{15}$N, ±0.25 ‰ for $\delta^{18}$O, and ±0.10 ‰ for $\Delta^{17}$O, respectively. Thank you for your advising. We would like to clarify this in the revised manuscript.

> Line 226: How was the error range "allowed"?

We estimated the uncertainty derived from the difference in the locality as 1 ‰. This was based on the standard deviation between the annual average $\Delta^{17}$O values determined in four different monitoring stations located in the same mid-latitudes, in the past studies such as La Jolla (33° N; Michalski et al., 2003), Princeton (40° N; Kaiser et al., 2007), Rishiri (45° N; Tsunogai et al., 2010), and Sado (38° N; Tsunogai et al., 2016). Besides, we estimated the uncertainty derived from the seasonal difference in the $\Delta^{17}$O values of atmospheric nitrate as 1.8 ‰, based on the standard deviation of six-month moving averages of atmospheric nitrate determined at the Sado monitoring station. Adding an additional 0.2 ‰ as a margin, we adopted 3 ‰ as the possible error for $\Delta^{17}$O atm in the streams.

> Line 279: I believe "events" should be singular.

Thank you for your advising. We would like to revise as suggested.

> Line 353: I suggest inserting "primarily" or "likely" before "responsible" here and elsewhere that this conclusion is presented.   The soil and stream data the authors are using come from different years as they describe on lines 318-342, so I think the conclusion on lines 351-354 should be considered tentative.

Thank you for your advising. We would like to revise as suggested.

> Lines 389-390: Please indicate which symbols indicate upland samples and which indicate riparian samples.

Thank you for your advising. We would like to revise as suggested.

We would like to thank you for the helpful comments and suggestions. We trust that our responses to your comments and questions are satisfactory.

Sincerely,
Weitian Ding

Reference

Bostic, J. T., Nelson, D. M., Sabo, R. D. and Eshleman, K. N.: Terrestrial Nitrogen Inputs Affect the Export of Unprocessed Atmospheric Nitrate to Surface Waters: Insights from Triple Oxygen Isotopes of Nitrate, Ecosystems, doi:10.1007/s10021-021-00722-9, 2021.

Christopher, S.F., Mitchell, M.J., McHale, M.R., Boyer, E.W., Burns, D.A. and Kendall, C. (2008), Factors controlling nitrogen release from two forested catchments with contrasting hydrochemical responses. Hydrol. Process., 22: 46-62. https://doi.org/10.1002/hyp.6632

Creed, I. F., Band, L. E., Foster, N. W., Morrison, I. K., Nicolson, J. A., Semkin, R. S. and Jeffries, D. S.: Regulation of nitrate-N release from temperate forests: A test of the N flushing hypothesis, Water Resour. Res., 32(11), 3337–3354, doi:10.1029/96WR02399, 1996.

Kaiser, J., Hastings, M. G., Houlton, B. Z., Röckmann, T. and Sigman, D. M.: Triple oxygen isotope analysis of nitrate using the denitrifier method and thermal decomposition of N2O, Anal. Chem., 79(2), 599–607, doi:10.1021/ac061022s, 2007.

Kamisako, M., Sase, H., Matsui, T., Suzuki, H., Takahashi, A., Oida, T., Nakata, M., Totsuka, T. and Ueda, H.: Seasonal and annual fluxes of inorganic constituents in a small catchment of a Japanese cedar forest near the sea of Japan, Water. Air. Soil Pollut., 195(1–4), 51–61, doi:10.1007/s11270-008-9726-8, 2008.

Kotlash, A. R. and Chessman, B. C.: Effects of water sample preservation and storage on nitrogen and phosphorus determinations: Implications for the use of automated sampling equipment, Water Res., 32(12), 3731–3737, doi:10.1016/S0043-1354(98)00145-6, 1998.

Michalski, G., Scott, Z., Kabiling, M. and Thiemens, M. H.: First measurements and modeling of Δ17O in atmospheric nitrate, Geophys. Res. Lett., 30(16), 3–6, doi:10.1029/2003GL017015, 2003.

Nakagawa, F., Tsunogai, U., Obata, Y., Ando, K., Yamashita, N., Saito, T., Uchiyama, S., Morohashi, M. and Sase, H.: Export flux of unprocessed atmospheric nitrate from temperate forested catchments: A possible new index for nitrogen saturation, Biogeosciences, 15(22), 7025–7042, doi:10.5194/bg-15-7025-2018, 2018.

Shanley, J. B., McDowell, W. H. and Stallard, R. F.: Long-term patterns and short-term dynamics of stream solutes and suspended sediment in a rapidly weathering tropical watershed, Water Resour. Res., 47(7), 1–11, doi:10.1029/2010WR009788, 2011.

Tsunogai, U., Komatsu, D. D., Daita, S., Kazemi, G. A., Nakagawa, F., Noguchi, I. and Zhang, J.: Tracing the fate of atmospheric nitrate deposited onto a forest ecosystem in Eastern Asia using Δ17O, Atmos. Chem. Phys., 10(4), 1809–1820, doi:10.5194/acp-10-1809-2010, 2010.

Tsunogai, U., Miyauchi, T., Ohyama, T., Komatsu, D. D., Nakagawa, F., Obata, Y., Sato, K. and Ohizumi, T.: Accurate and precise quantification of atmospheric nitrate in streams draining land of various uses by using triple oxygen isotopes as tracers, Biogeosciences, 13(11), 3441–3459, doi:10.5194/bg-13-3441-2016, 2016.

---

## Author Comment (AC2)

Reply to your comment (Referee #2).

Thank you very much for your valuable comments on our manuscript. We have responded to each of your comments and questions one by one.

> it is not clear to me how the "stable" "unprocessed" atmospheric nitrate can be used to evaluate nitrogen saturation in forested catchments.

Nakagawa et al. (2018) lately proposed that the $M_{atm}/D_{atm}$ ratio, the export flux of $NO_3^-{}_{atm}$ ($M_{atm}$) relative to the deposition flux of $NO_3^-{}_{atm}$ ($D_{atm}$), can be an alternative, more robust index to evaluate nitrogen saturation in each forested catchment, because the $M_{atm}/D_{atm}$ ratio directly reflect the demand on atmospheric nitrate deposited onto each forested catchments as a whole, and thus reflect the nitrogen saturation in each forested catchment. If the forested catchments under the nitrogen saturation, the demand on atmospheric nitrate of the forested catchments will decrease, and the export flux of $NO_3^-{}_{atm}$ ($M_{atm}$) will increase. Also, because $D_{atm}$ is variable between the different forested catchments, normalizing $M_{atm}$ by $D_{atm}$ is necessary for compare $M_{atm}$ between the different forested catchments. We would like to emphasize this in the revised MS.

> I'm also not able to follow why the conclusion of "the storm events have little impacts on the concentration of unprocessed atmospheric nitrate in the stream" is important and how the conclusion is arrived.

The concentrations of the unprocessed atmospheric nitrate ($[NO_3^-{}_{atm}]$) in rainwater (the $[NO_3^-{}_{atm}]$ in rainwater determined in Sado island, for example, was $27.2 \pm 18.5$ µM from August to October in 2009, 2010, and 2011; Tsunogai et al., 2016) were much higher than those in stream water ($2.2 \pm 0.6$ µM in this study). If significant portion of rainwater was added directly into the stream water during storm events, the $[NO_3^-{}_{atm}]$ in stream water should increase. The $[NO_3^-{}_{atm}]$ in stream water, however, was stable having no linear relationship with the precipitation or the total concentration of the stream nitrate during the storm events. As a result, we concluded that the directed input of the $[NO_3^-{}_{atm}]$ into the stream water was negligible even during the storm events. In addition, we also concluded that the $M_{atm}/D_{atm}$ ratio is controlled by the nitrogen saturation stage in each forest. Instead of direct input into the stream water during storm events, the $NO_3^-{}_{atm}$ experiences the metabolized processes (uptake or denitrification) in forested catchment subsequent to deposition, indicating that the $M_{atm}/D_{atm}$ ratio reflect the total demand on $NO_3^-{}_{atm}$ in each forested catchment and thus the nitrogen saturation status. We would like to clarify this in the revised MS.

> Overall, I'm not able to follow why "unprocessed atmospheric nitrate fraction" in river water is so important that the authors have to repeat and emphasize many times in the manuscript.

We mentioned the concentration of the unprocessed atmospheric nitrate ($[NO_3^-{}_{atm}]$) several times in the MS. The $M_{atm}/D_{atm}$ ratio can be controlled by two factors in forested catchments, (1) the hydrologic flow path, (2) nitrogen saturation stage. To verify the $M_{atm}/D_{atm}$ ratio can reflect the nitrogen saturation stage of the forested catchments, the amount of the direct input of the atmospheric nitrate in rainwater during storm events should be clarify. As a result, we discussed the amount of the direct input of the atmospheric nitrate in rainwater during storm events precisely and mentioned the $[NO_3^-{}_{atm}]$ in stream water many times.

> My understanding is that with finite fraction of atmospheric nitrate, one can utilize the unique triple oxygen isotope composition in atmospheric nitrate for riverine nitrogen dynamics study, which is what the group did in the past years. The fraction of "unprocessed atmospheric nitrate" represents a balance of release of soil nitrate and atmospheric deposition.

While the fraction of "unprocessed atmospheric nitrate" represents a balance of release of soil nitrate and atmospheric deposition in past studies (Nakagawa et al., 2018), it turns out that, while the concentration of stream nitrate increased, the $[NO_3^-{}_{atm}]$ in stream water remained almost stable during the storm events in this study indicating that the fraction of "unprocessed atmospheric nitrate" can't represent a balance of release of soil nitrate and atmospheric deposition. Thank you for your advice.

> Line 25-30: no flux estimation is provided, and so it is not clear how the statement of "the annual export flux of unprocessed atmospheric nitrate relative to the annual deposition flux" is obtained. Overall, from my understanding, the value of NO3_atm is quite stable. The values of the 3 storms are 1.6+/-0.4, 1.8+/-0.4, and 2.1+/-0.4 uM, while that during non-storm time is 2.2+/-0.6 uM. Isn't it more valuable to discuss storm and non-storm samples in the same context of nitrogen saturation and dynamics?

The annual export flux of unprocessed atmospheric nitrate relative to the annual deposition flux ($M_{atm}/D_{atm}$ ratio) was estimated from annual $[NO_3^-{}_{atm}]$ in the stream, annual flow rate of stream, and annual deposition flux of atmospheric nitrate. In the forested catchment, the annual flow rate of stream and annual deposition flux of atmospheric nitrate can be considered as constant. The $[NO_3^-{}_{atm}]$ in the stream was $1.6 \pm 0.4$ μM, $1.8 \pm 0.4$ μM, and $2.1 \pm 0.4$ μM during the storm events I, II, and III, respectively, which have no significant difference with the annual $[NO_3^-{}_{atm}]$ in the

stream ($2.2 \pm 0.6$ µM). Thus, the storm events have little impacts on the $M_{atm}/D_{atm}$ ratio. Thank you for your advising. We would like to clarify this in the revised MS.

> The term "enriched" may cause confusion. In isotope community, often the term is used for indicating an increase in isotope values, i.e., increase in the abundance of heavier isotopic compounds.

Thank you for your advising. We would like to revise the title as "Tracing the source of nitrate in a forested stream showing elevated concentrations during storm events" in the revised manuscript.

> Line 121: M_atm, D_atm are not defined till much later in section 4.3. Even in section 4.3, the two variables are not clearly defined and explained. Instead, the authors referred to their earlier paper (Nakagawa et al., 2018). The authors are fine to have the details in their previous paper but the authors have to at least explain the meaning of the two.

Thank you for your advising. We would like to revise as suggested.

> M_atm (or NO3_atm) is obtained by assuming a certain number of D17O_atm, which is not measured in this work. And so, D_atm is not known. Please elaborate and explain why M_atm/D_atm is little affected by storms and how this conclusion is arrived.

We have mentioned that above.

> Line 163: Please discuss whether 1-2 weeks of storage would affect the sample nitrate concentration and isotope compositions.

We think the microbial alteration of the stream water samples during the storage period can be negligible.
(1) The sampler was set on the riverbank near a weir surrounded by ferns and other understory vegetation avoiding sunlight during the observation. In addition, the bottles are stored in a shaded space to minimize the microbial alteration of the samples. Besides, the automatic sampler (SIGMA 900, Hach, USA) has equipped with refrigerator to keep the samples in 4°C. (2) Kotlash and Chessman (1998) have assessed the storage effects of freezing, acidification, refrigeration and extended storage without refrigeration (6 days) on measured concentrations of nitrogen of different stream water samples, and found there     was little difference in concentration of oxidized nitrogen ($NO_3^- + NO_2^-$) according to different treatment. (3) The concentrations of stream nitrate showed temporal variation in accordance with the variation in the stream flow rate during storm events (Figs. 3 and S1). As a result, the

variation of the stream nitrate concentrations was primarily controlled by the flow rate instead of the microbial process. (4) The $\Delta^{17}O$ of stream nitrate is stable during the progress of such microbial processes (e.g., denitrification or assimilation). While the $\delta^{15}N$ and $\delta^{18}O$ of stream nitrate can be altered by the progress of partial removal through microbial process, the $\delta^{15}N$ and $\delta^{18}O$ of stream nitrate showed strong linear relationship between the reciprocal of concentrations, implying that the primary process controlling both $\delta^{15}N$ and $\delta^{18}O$ was mixing. Thank you for your advising. We would like to emphasize this in the revised manuscript.

> Line 428, enhancement of D17O on 2019/1/31: I did a simple estimate by assuming that the snow nitrate has the same D17O value as the atmospheric at 26 per mil and took 2018/12/28 as an initial state before snow melting. From 2018/12/28 to 2019/1/31, the D17O value increases by 7 per mil, implying ~30% (=7 per mil/26 per mil) of stream nitrate is from snow melting. This increase however is not reflected in the water flow rate (from 110.0 to 117.3 L/min only). Please elaborate and provide a more quantitative explanation.

From 2018/12/28 to 2019/1/31, the $\Delta^{17}O$ value doesn't increase by 7 ‰, by +2.73 ‰ instead (Table S1). The flow rate, concentration of stream nitrate, and $\Delta^{17}O$ was 110.0 L/min, 70.0 μM, and +1.17 ‰ on 2018/12/28, respectively, and 117.3 L/min, 62.4 μM, and +2.73 ‰ on 2019/1/31, respectively. The $[NO_3^-{}_{atm}]$ in stream water was estimated to be 3.1 μM on 2018/12/28 and 6.5 μM on 2019/1/31. Assuming the $[NO_3^-{}_{atm}]$ in snow melt to be the same with that in rainwater (87.8 μM in the $[NO_3^-{}_{atm}]$ in rainwater at Sado island in January from 2009 to 2011; Tsunogai et al., 2016), we can estimate the amount of melting snow water into the stream water to be 4.8 L/min by using the mass balance law ($[NO_3^-{}_{atm}]_{2019/1/31}$ * flow$_{2019/1/31}$ = $[NO_3^-{}_{atm}]_{2018/12/28}$ * flow$_{2018/12/28}$ + $[NO_3^-{}_{atm}]_{snowmelt}$ * flow$_{snowmelt}$). As a result, the estimated amount of melting snow water into the stream water (4.8 L/min) less than the increase of flow rate (7.3 L/min; 117.3 L/min - 110.0 L/min), proved that the increased $\Delta^{17}O$ on 2019/1/31 can caused by the snow melting. Thank you for your advising. We would like to emphasize this in the revised MS.

> To be more complete, for routine sampling analysis and discussion, please include precipitation and do the same analysis as the storm events.

Thank you for your advising. We would like to add the data and discuss the data as suggested.

> Fig 4: it seems there are two groups (one having smaller slope and one steeper) of D17O vs. 1/[NO3-] in the storm event II. Any reason for that?

The increase of $\Delta^{17}O$ (steeper groups) could be caused by the input of the small amount of the $NO_3^-{}_{atm}$ in rainwater during the storm event II. Anyway, in the storm event II, the $\Delta^{17}O$ of stream nitrate showed strong linear relationship ($R^2$=0.81; P<0.0001) between the reciprocal of concentrations as whole, further, the $\Delta^{17}O$ of the riparian soil nitrate were plotted on the extension line indicated the primarily source of stream nitrate increased during storm event II was also riparian soil nitrate instead of the $NO_3^-{}_{atm}$ in rainwater. Thank you for your advising.

We would like to thank you for the helpful comments and suggestions. We trust that our responses to your comments and questions are satisfactory.

Sincerely,
Weitian Ding

Reference

Kotlash, A. R. and Chessman, B. C.: Effects of water sample preservation and storage on nitrogen and phosphorus determinations: Implications for the use of automated sampling equipment, Water Res., 32(12), 3731–3737, doi:10.1016/S0043-1354(98)00145-6, 1998.

Nakagawa, F., Tsunogai, U., Obata, Y., Ando, K., Yamashita, N., Saito, T., Uchiyama, S., Morohashi, M. and Sase, H.: Export flux of unprocessed atmospheric nitrate from temperate forested catchments: A possible new index for nitrogen saturation, Biogeosciences, 15(22), 7025–7042, doi:10.5194/bg-15-7025-2018, 2018.

Tsunogai, U., Miyauchi, T., Ohyama, T., Komatsu, D. D., Nakagawa, F., Obata, Y., Sato, K. and Ohizumi, T.: Accurate and precise quantification of atmospheric nitrate in streams draining land of various uses by using triple oxygen isotopes as tracers, Biogeosciences, 13(11), 3441–3459, doi:10.5194/bg-13-3441-2016, 2016.

---

## Author Response (AR1)

14 June 2022

Dr. Perran Cook
Editor of Biogeosciences

Title: Tracing the source of nitrate enriched in a forested stream during storm events
Authors: Weitian Ding et al.
MS No.: bg-2022-30

Dear Dr. Cook:

Thank you very much for handling our manuscript. We would like to thank the referees as well for the constructive comments on our manuscript. We have carefully studied the comments and revised the manuscript accordingly. We include below point-by-point responses to the comments, and detailed descriptions of the modifications we made to the manuscript. Besides, we also uploaded the revised manuscript in MS Word, in which all the revisions from BGD version were recorded. We hope that with these changes you will find our revised manuscript appropriate for publication in your journal.

Sincerely yours,
Weitian Ding
PhD student
Graduate School of Environmental Studies,
Nagoya University
Furo-cho, Chikusa-ku, Nagoya,
464-8601, JAPAN
Phone: +81-70-4436-3157
E-mail: ding.weitian.v2@s.mail.nagoya-u.ac.jp

**Response to the handing associate editor:**

> **I find the storage and of unfiltered and unrefrigerated samples for a period of weeks highly problematic and I am inclined to reject the manuscript on this basis. To my mind your observations could be explained by increased particulate material which occurs during high flow events being mineralised during storage leading to higher nitrate concentrations during this period. I will however give you a chance to respond to this major issue. I note that you have also taken samples that were immediately returned to the lab and processed, but it is unclear which samples these were in the data you presented. Can you separate out these samples to give us some confidence that there is no difference between the autosampler samples and grab samples?**

Thank you for the comment and the advising. We added the section of 4.1 and Appendix in the revised MS (P20/L308-324 and P35-39/L572-664) to discuss the possible alterations to the concentration and isotopic compositions of stream nitrate during the storage period in the automatic sampler used for the intensive observations.

Firstly, we added the specific storage information of the samples taken during the intensive observation (Table 1). We compared the samples taken during the intensive observations using the automatic sampler with those taken during the routine observations and found they coincided well each other (Table A1), implying that at least the progress of nitrification within the bottles should be minor.

In addition, a clear storm event was also observed during the routine observation on 2018/8/31, so that we can compare the concentrations and isotopic compositions of stream nitrate with those of intensive observations. Accompany with the precipitation and increase of flow rate on 2018/8/31, the trend and the degree of the variations in the concentration and the isotopic compositions from those on one month before were consistent with those of the intensive observation. Thus, the variation of the concentration of stream nitrate during storm event were controlled by the variation of the flow rate (flushing effect), instead of the progress of the nitrification during the storage period in the automatic sampler. Further, the similar increase in the concentrations of stream nitrate in accordance with the increase in the flow rate during storm events in past studies was cited in the revised MS (P17/L282-291), which also implied the increase of the concentration of stream nitrate was controlled by the flow rate (flushing effect).

Besides, Kotlash and Chessman (1998) conducted storage experiments under various conditions such as freezing, acidification, refrigeration, and room temperature to clarify the changes in the concentrations of nitrogen compounds in stream water samples and found little change in concentration of oxidized nitrogen ($NO_3^- + NO_2^-$) irrespective of the treatments. To further verify the insignificant changes in the concentrations and isotopic compositions of stream nitrate stored without treatments in the samples taken by the automatic sampler, we also conducted the storage

experiments by using a 100 mL of stream water taken at the KJ forested catchment on 2022/4/28. The concentration of nitrate in the stream water sample being stored for 2 weeks without treatments coincided well with those in the original, showing the difference in concentrations less than 10 % and the differences in the isotopic compositions from the original were also negligibly small (Table A2). As a result, we concluded that the possible alteration in the concentration and isotopic compositions of nitrate due to the progress of biogeochemical reactions such as nitrification, denitrification, and assimilation during storage in the automatic sampler used in the intensive observations was negligibly small.

Lastly, we mentioned that the observed strong linear relationships not only in the $\Delta^{17}O$ of stream nitrate, which is stable during the progress of partial removal reactions such as denitrification or assimilation, but also in the $\delta^{15}N$ and $\delta^{18}O$ of stream nitrate, which should be altered during the progress of the partial removal reactions, also implied that the progress of denitrification or assimilation in bottles of the automatic sampler during the storage period without filtration were minor in the samples (P21/L337-343).

Kotlash, A. R. and Chessman, B. C.: Effects of water sample preservation and storage on nitrogen and phosphorus determinations: Implications for the use of automated sampling equipment, Water Res., 32(12), 3731–3737, doi:10.1016/S0043-1354(98)00145-6, 1998.

> **There is no interpretation of your observed decrease in d15N during the storm events. Normally, I would expect to see an increase in d15N during flow events as the saturation of soil induces anoxic and denitrification which will increase d15N. Additionally it will stimulate the inflow of shallow groundwater which is also enriched in d15N. See for example https://bg.copernicus.org/articles/15/3953/2018/ (and many other refs, so feel no obligation to cite this). To my mind, the drop in d15N suggests the input of freshly fixed nitrogen and the low d15N of soil nitrate seems consistent with this. Is there evidence for nitrogen fixing plants in this catchment? I think a brief discussion of this would be worthwhile.**

Thank you for the comment and advising. We added a discussion in the revised MS to interpret the observed decrease in $\delta^{15}N$ of stream nitrate during the storm events (P25/L415-430).

In briefly, past studies have reported significant differences between the $\delta^{15}N$ values of soil nitrate and those of stream nitrate in six forested catchments in Japan and China, and proposed that the kinetic fractionation due to the progress of denitrification during the elution of soil nitrate into groundwater was responsible for the relative [15]N-enrichment in stream nitrate compared with soil nitrate (Fang et al., 2015; Hattori et al., 2019). In this study, compared with the $\delta^{15}N$ values of stream

nitrate taken during the base flow periods of routine observations, the riparian soil nitrate showed the $\delta^{15}N$ values around 3.5 ‰ lower. The trend and the extent of the $^{15}N$-depletion coincided well with those determined in the forested catchments in past studies (Fang et al., 2015; Hattori et al., 2019). As a result, the observed temporal decrease in the $\delta^{15}N$ value of stream nitrate during storm events also supported that the flushing of soil nitrate showing $^{15}N$-depleted $\delta^{15}N$ values into the stream was responsible for the elevated of nitrate concentrations during storm events.

Hattori, S., Nuñez Palma, Y., Itoh, Y., Kawasaki, M., Fujihara, Y., Takase, K. and Yoshida, N.: Isotopic evidence for seasonality of microbial internal nitrogen cycles in a temperate forested catchment with heavy snowfall, Sci. Total Environ., 690, 290–299, doi:10.1016/j.scitotenv.2019.06.507, 2019.
Fang, Y., Koba, K., Makabe, A., Takahashi, C., Zhu, W., Hayashi, T., Hokari, A. A., Urakawa, R., Bai, E., Houlton, B. Z., Xi, D., Zhang, S., Matsushita, K., Tu, Y., Liu, D., Zhu, F., Wang, Z., Zhou, G., Chen, D., Makita, T., Toda, H., Liu, X., Chen, Q., Zhang, D., Li, Y. and Yoh, M.: Microbial denitrification dominates nitrate losses from forest ecosystems, Proc. Natl. Acad. Sci. U. S. A., 112(5), 1470–1474, doi:10.1073/pnas.1416776112, 2015.

**> I suggest you delete the 1000 in front of d15N etc**

Thank you for the suggestion. We have revised that in Figs 2, 3, 4, 5, and S1 as suggestion.

**> Please label your time series x axes and specify unit (time, month etc).**

Thank you for the suggestion. We have revised that in Figs 2, 3, 5, and S1 as suggestion.

**Response to the referee #1:**

> **For example, Sebestyen et al. 2019 (ES&T) and the references it contains address a similar issue as this manuscript (e.g. Buda et al. 2009 and Sabo et al. 2016 both sampled storm events), and this manuscript could do a more throughout job of using those studies to help justify this study (in the Introduction) and then comparing/contrasting the results of this study to those studies in the Discussion. Similarly, oher studies (Burns et al. 2009; Barnes et al. 2010; Bostic et al. 2021) have addressed similar questions in non-forested systems and could be useful for helping to provide a broader context for the results that are presented in this manuscript.**

We cited many of the suggested past studies in the introduction and compared the results of them in the discussion in the revised MS (P5-6/L81-85 and P22/L353-357).

> **Title: "Enriched" is a word that is often used incorrectly in the isotope literature to refer to increased values of the heavier isotope. Here I believe the authors use "enriched" to mean increased nitrate concentrations, which is might cause confusion given that this paper also talks about isotopic enrichment (e.g lines 66 and 303). One solution might be to simply delete "enriched" from the title and another solution might be replace "nitrate enriched" in the title with something like "elevated nitrate concentrations"**

We revised the title as "Tracing the source of nitrate in a forested stream showing elevated concentrations during storm events" in the revised manuscript.

> **Lines 2-3: This sentence implies that nitrate concentrations always increase in temperate forest streams everywhere. Is that true? If not, perhaps slightly adjust this sentence.**

While many past studies reported increasing the stream nitrate concentration during storm events in temperate forest (e.g. Creed et al., 1996; Kamisako et al., 2008; Christopher., 2008), the decrease pattern (Christopher., 2008) or stable pattern (Shanley et al., 2011) of stream nitrate concentration during storm events also have been reported. We revised the sentence as suggested (P2/L2).

> **For example, do some severely nitrogen saturated forests that show higher NO3 concentrations in baseflow than stormflow?**

No. For example, the KJ forested catchment has been reported under severely nitrogen saturation (Nakagawa et al., 2018), both this study and past study (Kamisako

et al., 2008) found significant increase in the stream nitrate concentration during stormflow than baseflow.

**> Line 5: Please tell the reader what time of year (winter, spring, summer, autumn) these storm events occurred.**

We revised the sentence as suggested (P2/L5).

**> Line 6: It might be helpful to insert "increasing" before "from" to help the reader understand that the "variation" nitrate concentration that was observed was primarily an increase in concentrations.**

We revised the sentence as suggested (P2/L6).

**> Line 14: I believe "($d^{15}N$, $d^{18}O$, and $C^{17}O$)" can be deleted without sacrificing meaning.**

We revised the sentence as suggested (P2/L14).

**> Line 26-27: Could the authors support this claim by calculating annual export of NO3-atm (and NO3-terr) using their concentration and flow data?**

The annual export of unprocessed atmospheric nitrate in the stream can be calculated as $3.2 \pm 0.7$ mmol m$^{-2}$ yr$^{-1}$ by multiplying the average flow rate of stream and the average concentration of unprocessed atmospheric nitrate in the stream during the routine observation.

The annual export flux of unprocessed atmospheric nitrate relative to the annual deposition flux ($M_{atm}/D_{atm}$ ratio), nitrogen saturation index, was estimated from annual concentration of unprocessed atmospheric nitrate in the stream, annual flow rate of stream, and annual deposition flux of atmospheric nitrate. In the forested catchment, the annual flow rate of stream and annual deposition flux of atmospheric nitrate can be considered as constant. The concentration of unprocessed atmospheric nitrate in the stream was $1.6 \pm 0.4$ µM, $1.8 \pm 0.4$ µM, and $2.1 \pm 0.4$ µM during the storm events I, II, and III, respectively, which have no significant difference with the average concentration of unprocessed atmospheric nitrate in the stream ($2.2 \pm 0.6$ µM). Thus, the storm events have little impacts on the $M_{atm}/D_{atm}$ ratio. We emphasized this in the revised MS (P33/L532-544).

**> Lines 26-30: Is this conclusion specific to the author's study site (or certain types of forests) or are they suggesting that is a more broad/general conclusion that applies to forested catchments everywhere?**

We conducted the research at KJ forested catchments as an example of the nitrogen saturated forested catchment. The conclusion is only suitable for the KJ forested catchment at present. Further works should be needed to verify the conclusion in different forested catchments in the future.

**> Line 33: "representative" of what?   Please clarify.**

We revised the sentence in the revised MS (P3/L33).

**> Line 50: First, how are the authors using "overland flow" here and elsewhere (e.g. line 463) in the manuscript?   My understanding is that overland flow is unlikely in areas that are not near channels or stream/riparian areas in forests except for unique situations, such as intense rain events or rain that occurs on frozen soils. Second, I don't believe either of the cited studies suggest that overland flow is a mechanism for direct suppler of atmospheric nitrate to stream water. As far as I recall, Kaushal et al. didn't show overland flow for their forested site and Sebestyen et al. talked about routing of NO3-atm along flow paths that allowed NO3-atm to bypass uptake/processing (but not specifically about overland flow).**

We revised the sentence and cited a new article in the revised MS (P4/L49-50).

Inamdar, S. P. and Mitchell, M. J.: Hydrologic and topographic controls on storm-event exports of dissolved organic carbon (BOC) and nitrate across catchment scales, Water Resour. Res., 42(3), 1–16, doi:10.1029/2005WR004212, 2006.

**> Line 72: Is beta completely constant or can it exhibit some variation around 0.5279? If so, does the variation affect the authors data analyses or interpretations?**

Bostic et al. (2021) assumed the $\beta$ could vary from 0.51 to 0.53. In the whole samples analyzed in this study (n=105), both the min and max value of the $\delta^{18}O$ was -3.3 ‰ and +7.7 ‰, respectively. Thus, the max range of deviation in the $\Delta^{17}O$ could be estimated to be 0.1 ‰ (Fig 1), which in accordance with our analytical standard error of $\Delta^{17}O$. As a result, our conclusion cannot be influenced by the variation of $\beta$.

[Figure]

Figure 1. Schematic diagram showing the max variation of the $\Delta^{17}O$ in accordance with the variation of the $\beta$ from 0.51 to 0.53 in this study.

**> Lines 162-164: It seems like there would be potential for microbial alteration of the samples during the 1-2 weeks that they stayed in the field before being returned to the lab. Did the authors assess this?**

We answered the question in the above.

**> Lines 184-185: How many "local laboratory nitrate standards" were used and what are their isotope values?**
**> Lines 205-206: What data were used to calculate the reported standard error of the mean for each isotope? For example, was precision determined from the lab standards, replicate samples, or something else?**

In this study, we used three kinds of the local laboratory nitrate standards, which were named to be GG01 ($\delta^{15}N = -3.07$ ‰, $\delta^{18}O = +1.10$ ‰, and $\Delta^{17}O = 0$ ‰), HDLW02 ($\delta^{15}N = +16.11$ ‰, $\delta^{18}O = +22.20$ ‰), and NF ($\Delta^{17}O = +19.16$ ‰), which the GG01 and the HDLW02 were used to determine the $\delta^{15}N$ and $\delta^{18}O$ of stream nitrate, and the GG01 and the NF was used to determine the $\Delta^{17}O$ of stream nitrate. The standard error of the mean of the isotopic compositions ($\delta^{15}N$, $\delta^{18}O$, and $\Delta^{17}O$) was determined by repeated measurements of the GG01 (n = 3), ±0.17 ‰ for $\delta^{15}N$, ±0.25 ‰ for $\delta^{18}O$, and ±0.10 ‰ for $\Delta^{17}O$, respectively. We added the sentence in the revised MS (P12/L206-214)

**> Line 226: How was the error range "allowed"?**

We estimated the uncertainty derived from the difference in the locality as 1 ‰. This was based on the standard deviation between the annual average $\Delta^{17}O$ values determined in four different monitoring stations located in the same mid-latitudes, in the past studies such as La Jolla (33° N; Michalski et al., 2003), Princeton (40° N; Kaiser et al., 2007), Rishiri (45° N; Tsunogai et al., 2010), and Sado (38° N; Tsunogai et al., 2016). Besides, we estimated the uncertainty derived from the seasonal difference in the $\Delta^{17}O$ values of atmospheric nitrate as 1.8 ‰, based on the standard deviation of six-month moving averages of atmospheric nitrate determined at the Sado monitoring station. Adding an additional 0.2 ‰ as a margin, we adopted 3 ‰ as the possible error for $\Delta^{17}O$ atm in the streams.

**> Line 279: I believe "events" should be singular.**

We revised the sentence in the revised MS (P19/L302).

**> Line 353: I suggest inserting "primarily" or "likely" before "responsible" here and elsewhere that this conclusion is presented. The soil and stream data the authors are using come from different years as they describe on lines 318-342, so I think the conclusion on lines 351-354 should be considered tentative.**

We revised this in the revised MS (P2/L20, P24/L403, and P34/L562).

**> Lines 389-390: Please indicate which symbols indicate upland samples and which indicate riparian samples.**

We revised this in the revised MS (P27/L449-450).

**Response to the referee #2:**

**> it is not clear to me how the "stable" "unprocessed" atmospheric nitrate can be used to evaluate nitrogen saturation in forested catchments.**

Nakagawa et al. (2018) lately proposed that the $M_{atm}/D_{atm}$ ratio, the export flux of $NO_3^-{}_{atm}$ ($M_{atm}$) relative to the deposition flux of $NO_3^-{}_{atm}$ ($D_{atm}$), can be an alternative, more robust index to evaluate nitrogen saturation in each forested catchment, because the $M_{atm}/D_{atm}$ ratio directly reflect the demand on atmospheric nitrate deposited onto each forested catchments as a whole, and thus reflect the nitrogen saturation in each forested catchment. If the forested catchments under the nitrogen saturation, the demand on atmospheric nitrate of the forested catchments will decrease, and the export flux of $NO_3^-{}_{atm}$ ($M_{atm}$) will increase. Also, because $D_{atm}$ is variable between the different forested catchments, normalizing $M_{atm}$ by $D_{atm}$ is necessary for compare $M_{atm}$ between the different forested catchments. We emphasized this in the revised MS (P32/L525-528).

**> I'm also not able to follow why the conclusion of "the storm events have little impacts on the concentration of unprocessed atmospheric nitrate in the stream" is important and how the conclusion is arrived.**

The concentrations of atmospheric nitrate ($[NO_3^-{}_{atm}]$) in rainwater were much higher than those in stream water. While the volume-weighted mean $[NO_3^-{}_{atm}]$ in rainwater determined in Sado island from August to October, for example, was 15.2 ± 8.4 µM (EANET, 2010, 2011; Tsunogai et al., 2016), that in the stream water was 2.2 ± 0.6 µM in this study. As a result, the $[NO_3^-{}_{atm}]$ in stream water would increase, if significant portion of rainwater was added directly into the stream water during the storm events. The $[NO_3^-{}_{atm}]$ in stream water, however, was stable showing no correlation with the amount of precipitation or the concentration of stream nitrate during the storm events. As a result, we concluded that the directed input of the $[NO_3^-{}_{atm}]$ into the stream water was negligible even during the storm events. In addition, we also concluded that the $M_{atm}/D_{atm}$ ratio is controlled by the nitrogen saturation stage in each forest. Instead of direct input into the stream water during storm events, the $NO_3^-{}_{atm}$ experiences the metabolized processes (uptake or denitrification) in forested catchment subsequent to deposition, indicating that the $M_{atm}/D_{atm}$ ratio reflect the total demand on $NO_3^-{}_{atm}$ in each forested catchment and thus the nitrogen saturation status. We emphasized this in the revised MS (P33/L532-544 and P33-34/L548-554).

**> Overall, I'm not able to follow why "unprocessed atmospheric nitrate fraction" in river water is so important that the authors have to repeat and emphasize many times in the manuscript.**

We mentioned the concentration of the unprocessed atmospheric nitrate ($[NO_3^-{}_{atm}]$) several times in the MS. The $M_{atm}/D_{atm}$ ratio can be controlled by two factors in forested catchments, (1) the hydrologic flow path, (2) nitrogen saturation stage. To verify the $M_{atm}/D_{atm}$ ratio can reflect the nitrogen saturation stage of the forested catchments, the amount of the direct input of the atmospheric nitrate in rainwater during storm events should be clarify. As a result, we discussed the amount of the direct input of the atmospheric nitrate in rainwater during storm events precisely and mentioned the $[NO_3^-{}_{atm}]$ in stream water many times.

**> My understanding is that with finite fraction of atmospheric nitrate, one can utilize the unique triple oxygen isotope composition in atmospheric nitrate for riverine nitrogen dynamics study, which is what the group did in the past years. The fraction of "unprocessed atmospheric nitrate" represents a balance of release of soil nitrate and atmospheric deposition.**

While the fraction of "unprocessed atmospheric nitrate" represents a balance of release of soil nitrate and atmospheric deposition in past studies (Nakagawa et al., 2018), it turns out that, while the concentration of stream nitrate increased, the $[NO_3^-{}_{atm}]$ in stream water remained almost stable during the storm events in this study indicating that the fraction of "unprocessed atmospheric nitrate" can't represent a balance of release of soil nitrate and atmospheric deposition.

**> Line 25-30: no flux estimation is provided, and so it is not clear how the statement of "the annual export flux of unprocessed atmospheric nitrate relative to the annual deposition flux" is obtained. Overall, from my understanding, the value of NO3_atm is quite stable. The values of the 3 storms are 1.6+/-0.4, 1.8+/-0.4, and 2.1+/-0.4 uM, while that during non-storm time is 2.2+/-0.6 uM. Isn't it more valuable to discuss storm and non-storm samples in the same context of nitrogen saturation and dynamics?**

The annual export flux of unprocessed atmospheric nitrate relative to the annual deposition flux ($M_{atm}/D_{atm}$ ratio) was estimated from annual $[NO_3^-{}_{atm}]$ in the stream, annual flow rate of stream, and annual deposition flux of atmospheric nitrate. In the forested catchment, the annual flow rate of stream and annual deposition flux of atmospheric nitrate can be considered as constant. The $[NO_3^-{}_{atm}]$ in the stream was $1.6 \pm 0.4$ μM, $1.8 \pm 0.4$ μM, and $2.1 \pm 0.4$ μM during the storm events I, II, and III, respectively, which have no significant difference with the annual $[NO_3^-{}_{atm}]$ in the stream ($2.2 \pm 0.6$ μM). Thus, the storm events have little impacts on the $M_{atm}/D_{atm}$ ratio.

> **The term "enriched" may cause confusion. In isotope community, often the term is used for indicating an increase in isotope values, i.e., increase in the abundance of heavier isotopic compounds.**

We revised the title as "Tracing the source of nitrate in a forested stream showing elevated concentrations during storm events" in the revised manuscript.

> **Line 121: M_atm, D_atm are not defined till much later in section 4.3. Even in section 4.3, the two variables are not clearly defined and explained. Instead, the authors referred to their earlier paper (Nakagawa et al., 2018). The authors are fine to have the details in their previous paper but the authors have to at least explain the meaning of the two.**

We added a sentence to explain the $M_{atm}/D_{atm}$ ratio briefly (P7-8/L125-127).

> **M_atm (or NO3_atm) is obtained by assuming a certain number of D17O_atm, which is not measured in this work. And so, D_atm is not known. Please elaborate and explain why M_atm/D_atm is little affected by storms and how this conclusion is arrived.**

We answered the question in the above.

> **Line 163: Please discuss whether 1-2 weeks of storage would affect the sample nitrate concentration and isotope compositions.**

We answered the question in the above.

> **Line 428, enhancement of D17O on 2019/1/31: I did a simple estimate by assuming that the snow nitrate has the same D17O value as the atmospheric at 26 per mil and took 2018/12/28 as an initial state before snow melting. From 2018/12/28 to 2019/1/31, the D17O value increases by 7 per mil, implying ~30% (=7 per mil/26 per mil) of stream nitrate is from snow melting. This increase however is not reflected in the water flow rate (from 110.0 to 117.3 L/min only). Please elaborate and provide a more quantitative explanation.**

From 2018/12/28 to 2019/1/31, the $\Delta^{17}O$ value doesn't increase by 7 ‰, by +2.73 ‰ instead.

The flow rate, concentration of stream nitrate, and $\Delta^{17}O$ was 110.0 L/min, 70.0 μM, and +1.17 ‰ on 2018/12/28, respectively, while 117.3 L/min, 62.4 μM, and +2.73 ‰ on 2019/1/31, respectively. The $[NO_3^-{}_{atm}]$ in stream water was estimated to be 3.1 μM on 2018/12/28 and 6.5 μM on 2019/1/31. Assuming that the $[NO_3^-{}_{atm}]$ in snow melt was the same with the volume-weighted mean concentration of nitrate in rainwater

(41.0 μM) determined at Sado island in January (EANET, 2010, 2011; Tsunogai et al., 2016), the increase in the flow rate ($\Delta F_{snowmelt}$) due to the mixing of snow melt into the stream can be estimated to be 10.3 L/min, by using the mass balance equation shown below:

($[NO_3^-{}_{atm}]_{2019/1/31} \times F_{2019/1/31} = [NO_3^-{}_{atm}]_{2018/12/28} \times F_{2018/12/28} + [NO_3^-{}_{atm}]_{snowmelt} \times \Delta F_{snowmelt}$)

where $[NO_3^-{}_{atm}]_{2018/12/28}$, $[NO_3^-{}_{atm}]_{2019/1/31}$, and $[NO_3^-{}_{atm}]_{snowmelt}$ denote the $[NO_3^-{}_{atm}]$ in stream water on 2018/12/28, 2019/1/31, and that in snow melt water, respectively, and $F_{2018/12/28}$, $F_{2019/1/31}$, and $\Delta F_{snowmelt}$ denote the flow rate of stream water on 2018/12/28, 2019/1/31, and the increase in the flow rate due to snow melt, respectively. Because the estimated volume of melting snow water into the stream water (10.3 L/min) was comparable with the observed increase in the flow rate from 2018/12/28 to 2019/1/31 (7.3 L/min), we concluded that the snow melting was responsible for the increase in $\Delta^{17}O$ on 2019/1/31. We emphasized this in the revised MS (P31/L489-507).

> **To be more complete, for routine sampling analysis and discussion, please include precipitation and do the same analysis as the storm events.**

We added the precipitation in the table S1 and discussed it in the revised MS (P26 433-436).

> **Fig 4: it seems there are two groups (one having smaller slope and one steeper) of D17O vs. 1/[NO3-] in the storm event II. Any reason for that?**

The increase of $\Delta^{17}O$ (steeper groups) could be caused by the input of the small amount of the $NO_3^-{}_{atm}$ in rainwater during the storm event II. Anyway, in the storm event II, the $\Delta^{17}O$ of stream nitrate showed strong linear relationship ($R^2=0.81$; $P<0.0001$) between the reciprocal of concentrations as whole, further, the $\Delta^{17}O$ of the riparian soil nitrate were plotted on the extension line indicated the primarily source of stream nitrate increased during storm event II was also riparian soil nitrate instead of the $NO_3^-{}_{atm}$ in rainwater.